# Two missense mutations in *KCNQ1* cause pituitary hormone deficiency and maternally inherited gingival fibromatosis

Johanna Tommiska et al.[#]

Familial growth hormone deficiency provides an opportunity to identify new genetic causes of short stature. Here we combine linkage analysis with whole-genome resequencing in patients with growth hormone deficiency and maternally inherited gingival fibromatosis. We report that patients from three unrelated families harbor either of two missense mutations, c.347G>T p.(Arg116Leu) or c.1106C>T p.(Pro369Leu), in *KCNQ1*, a gene previously implicated in the long QT interval syndrome. *Kcnq1* is expressed in hypothalamic GHRH neurons and pituitary somatotropes. Co-expressing KCNQ1 with the KCNE2 β-subunit shows that both KCNQ1 mutants increase current levels in patch clamp analyses and are associated with reduced pituitary hormone secretion from AtT-20 cells. In conclusion, our results reveal a role for the KCNQ1 potassium channel in the regulation of human growth, and show that growth hormone deficiency associated with maternally inherited gingival fibromatosis is an allelic disorder with cardiac arrhythmia syndromes caused by *KCNQ1* mutations.

#A full list of authors and their affliations appears at the end of the paper

Somatic growth mirrors nutritional status, general health, and psychosocial well-being, and is also affected by inherited and epigenetic factors. An essential role in human growth is played by growth hormone which is secreted by the anterior pituitary somatotropes, mainly under the influence of hypothalamic inputs such as growth hormone-releasing hormone (GHRH) and somatostatin[1]. Childhood onset of growth hormone

deficiency (GHD) is a clinically heterogeneous condition and defining its cause is important for diagnostics and treatment.

Studying familial GHD provides an opportunity to identify genetic regulators of growth hormone secretion and pituitary function. The most common genes implicated in the genetic etiology of GHD are *GH1* (MIM: 139250, Online Mendelian Inheritance in Man, http://www.omim.org/), encoding growth

**Fig. 1** Pedigrees, gingival fibromatosis and craniofacial features, and KCNQ1 structure. **a** (I) Pedigree of the large Finnish family showing autosomal dominant growth hormone deficiency and maternally inherited gingival fibromatosis. The genotype (wild-type (WT) or p.(Arg116Leu)) is given. The samples included in the linkage analysis are indicated in italics, and samples included in whole-genome sequencing are underlined. Two additional families with the same disease but another mutation in *KCNQ1*, p.(Pro369Leu), were identified: a Finnish trio (II) and a family (III) originating from Argentina. Note that the index patient in pedigree III has a de novo mutation. **b** Maternally inherited gingival fibromatosis is shown together with the craniofacial features of the twin boys belonging to family II. **c** Schematic of the KCNQ1 channel protein and the location of the two missense mutations in the 3D channel structure. The schematic shows the membrane domain, with helical segments S0–S6 and the intracellular domain, divided into a membrane-proximal module (helices A–B) bound by CaM and a distal module (helices C–D), responsible for tetramerization. Filled circles with labels show the positions of the mutations, Arg116Leu and Pro369Leu. The double lines depict the plasma membrane. Below the schematic is a molecular graphics depiction of the Kv7.1/CaM channel complex as based on the cryo EM *Xenopus* structure (PDB code: 5VMS)[11]. The channel subunits are colored green, cyan, and teal. CaM is colored pink and shown with a surface representation on the right side. Gray spheres are Ca²⁺ ions. Again, the straight lines denote the probable location of the plasma membrane. The residues that undergo mutation are drawn as CPK atoms and are labeled. One channel subunit of the tetramer and its respective CaM molecule have not been drawn in order to facilitate visualization. The helix D tetrameric coiled-coil[62] was not observed in the cryo EM study due to flexibility in the linker between it and helix C and hence not drawn here

**Table 1 Summary of phenotypic features in patients with *KCNQ1* mutations p.(Arg116Leu) or p.(Pro369Leu) and pituitary hormone deficiencies**

| Subject | *KCNQ1* mutation | Sex | QTc interval in ECG (ms) | Height (SDS) at the age of the onset of GH therapy | Brain MRI | Pituitary hormone deficiencies | Mutation inherited/ gingival fibromatosis | Craniofacial phenotype as a child |
|---|---|---|---|---|---|---|---|---|
| #5 | p.R116L | F | 414 | −4.5 at 15 years | Normal | Growth hormone and gonadotropin | Maternally/Yes | NA |
| #6 | p.R116L | F | 412 | −3.4 at 12.4 years | Normal | Growth hormone and gonadotropin | Maternally/Yes | NA |
| #7 | p.R116L | M | 391 | −5.0 at 8.5 years | Small hypophysis | Growth hormone, gonadotropin, ACTH, and thyrotropin | Maternally/Yes | Yes |
| #13 | p.R116L | F | NA | −2.7 at 4.5 years | Normal | Growth hormone | Paternally/No | No |
| #13b | p.R116L | F | NA | −2.7 at 3.7 years | Normal | Growth hormone and thyrotropin | Paternally/No | No |
| #8 | p.R116L | M | 398 | −2.6 at 15.9 years | NA | Growth hormone and gonadotropin | Maternally/Yes | Yes |
| #9 | p.R116L | F | NA | −2.7 at 9 years | NA | Growth hormone, gonadotropin | Maternally/Yes | Yes |
| #15 | p.R116L | M | 363[a] | −1.8 at 6 years | Small hypophysis with thin stalk | Growth hormone, gonadotropin | Maternally/Yes | Yes |
| #17 | p.R116L | M | 329[a] | −2.2 at 5 years | Small hypophysis with thin stalk | Growth hormone, gonadotropin | Maternally/Yes | Yes |
| #18 | p.R116L | F | 463 | −2.3 at 13.4 years | Normal | Growth hormone | Maternally/Yes | Yes |
| #20 | p.P369L | F | 317[a] | −5.2 at 17 years | Normal | Growth hormone, gonadotropin, ACTH and thyrotropin | NA/No | No |
| #21 | p.P369L | M | 399 | −3.0 at 2.7 years | Small hypophysis | Growth hormone | Maternally/Yes | Yes |
| #22 | p.P369L | M | 358[a] | No GH therapy | NA | Growth hormone (no treatment) | Maternally/Yes | Yes |
| #25[b] | p.P369L | F | 349[a] | NA | Normal | No treatment | *De novo*/Yes | NA |

*F* female, *MRI* magnetic resonance imaging, *M* male, *NA* not available
[a]QTc time less than the 2nd percentile for gender and age[10,11]
[b]She has a de novo mutation. She has refused endocrine testing. Her two daughters have GF and craniofacial phenotype, and one of them also has microhypophysis and GH deficiency diagnosed at 5 years of age

hormone (GH), and *GHRHR* (MIM: 139191), encoding the receptor for GHRH[2]. GHD may also result from mutations in genes that encode transcription factors involved in pituitary development: *HESX1* (MIM: 601802), *OTX2* (MIM: 600037), *SOX2* (MIM: 184429), *SOX3* (MIM: 313430), *LHX3* (MIM: 600577), *PITX2* (MIM: 601542), *PROP1* (MIM: 601538), *POU1F1* (MIM: 173110), and *TCF7L1* (MIM:604652)[1,3]. Some of these mutations are associated with additional pituitary hormone deficiencies and developmental abnormalities, such as variants of septo-optic dysplasia (MIM: 182230), ocular defects, ectopic posterior pituitary, skeletal defects, and intellectual impairment[1].

Occurrence of growth retardation due to GHD in combination with gingival fibromatosis (GF), described hitherto in three families[4–6], suggests a previously unidentified genetic cause of GHD. Here we report that this syndrome is caused by two missense mutations, c.347G>T p.(Arg116Leu) or c.1106C>T p. (Pro369Leu), in *KCNQ1*, a gene that encodes the alpha subunit of the voltage-gated ion channel Kv7.1, previously implicated in cardiac arrhythmia syndromes[7–9]. We demonstrate the expression of *Kcnq1* in hypothalamic GHRH neurons and pituitary somatotropic cells. Both mutants increase current levels in patch clamp analyses, especially when co-expressed with the auxiliary KCNE2 β-subunit. Mutant KCNQ1s also lead to reduced pituitary hormone secretion from AtT-20 cells when coexpressed with KCNE2. Taken together, our results demonstrate that two missense mutations in *KCNQ1* cause pituitary hormone deficiency and maternally inherited GF in humans.

## Results

**Pedigree and genetic analysis**. We performed genome-wide linkage analysis in a large Finnish family[4] (Fig. 1a) with this disease, and found statistically significant linkage, logarithm of odds (LOD) score > 3.3, between the studied phenotype and a locus on 11p15. The highest LOD score, 3.54, was reached with the marker rs7947959 (hg19 chr11:4784679). The most distal marker showing statistically significant linkage was located at 286 kb and the most proximal marker at 11.36 Mb. No other chromosomal areas reached LOD scores >3.0. We then performed whole-genome resequencing (Beijing Genomics Institute) in two affected family members. Novel variants in the linkage region shared between the two genomes, and not found in any databases (dbSNP, http://www.ncbi.nlm.nih.gov/SNP/; Exome Variant Server, http://evs.gs.washington.edu/EVS/; 1000 Genomes, http://www.1000genomes.org/; ExAC, http://exac.broadinstitute.org/), were annotated with SNPnexus (http://www.snp-nexus.org/). The c.347G>T p.(Arg116Leu) missense mutation in *KCNQ1* (MIM: 607542), shared between the two genomes, was left in the linkage region after known variants, variants in the noncoding regions, and the synonymous changes were filtered out. PolyPhen-2 (http://genetics.bwh.harvard.edu/pph2/) predicted this mutation to be deleterious with very high probability (score 0.993). The presence or absence of the mutation was verified by Sanger-sequencing in all 16 family members, and in four additional affected family members participating in the study after completion of linkage analysis. The mutation segregated with GHD and/or short stature in the family and was

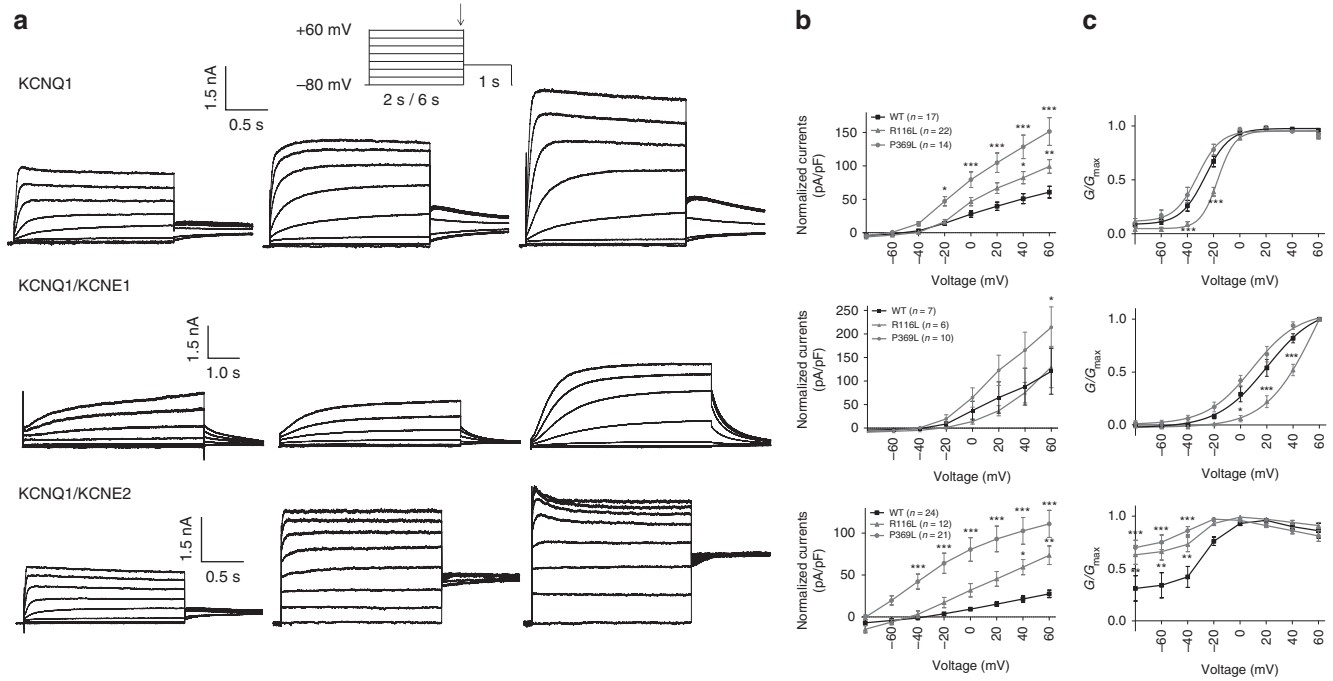

**Fig. 2** Electrophysiological studies of the WT, KCNQ1-Arg116Leu and KCNQ1-Pro369Leu mutant KCNQ1 proteins. **a** Representative recordings of currents measured during the voltage-clamp protocol (shown in inset with scale bars) in HEK293 cells transfected with cDNAs encoding wild-type (WT) or mutant (p.Arg116Leu, p.Pro369Leu) KCNQ1 potassium channels: KCNQ1 alone (top row), KCNQ1 and KCNE1 (middle row), or KCNQ1 and KCNE2 (bottom row). **b** Respective current-voltage relationships normalized for cell size. Peak current analysis was performed at the end of each voltage step. pA/pF: picoamperes per picofarad. **c** Normalized activation curves, measured 2–4 ms after stepping to −30 mV, as a function of the prior voltage potential. KCNQ1 and KCNQ1/KCNE1 conductance-voltage relation values were fitted to a Boltzmann function. The time constants and slope values were; KCNQ1 WT: −26.2 ± 2.9 mV, 8.6 ± 0.9 (n = 14/3), KCNQ1-R116L: −16.1 ± 1.5 mV, 6.9 ± 0.3 (n = 21/4), KCNQ1-P369L: −30.9 ± 2.5 mV, 6.9 ± 0.6 (n = 14/3), KCNQ1 WT+KCNE1: 21.0 ± 2.5 mV, 16.2 ± 1.7 (n = 18/3), KCNQ1-R116L+KCNE1: 42.4 ± 3.6 mV, 12.7 ± 1.7 (n = 11/2), KCNQ1-P369L+KCNE1: 7.3 ± 3.8 mV, 14.7 ± 0.6 (n = 18/3). Mean ± SEM values are shown. *P < .05, **P < .01, ***P < .001 vs. WT. G/G$_{max}$: conductance of the channel relative to its maximal conductance

absent from the SISu database (The Sequencing Initiative Suomi project, containing over 10,000 Finnish samples, http://www.sisuproject.fi/). Sanger sequencing of the coding exons and exon–intron boundaries of the *KCNQ1* gene in a second Finnish family (mother and twin boys) with GHD and GF revealed another novel missense mutation, c.1106C>T p.(Pro369Leu), in all three members (Fig. 1a, b). This mutation was also absent from the dbSNP, Exome Variant Server, 1000 Genomes, ExAC, and SISu databases and predicted to be probably damaging by PolyPhen-2 (score 0.960). In the third family, a Swiss family originating from Argentina (Fig. 1a), the index patient carried the same c.1106C>T p.(Pro369Leu) mutation as de novo.

**Phenotypic features of *KCNQ1* mutation carriers**. Detailed clinical histories of mutation carriers are provided in Supplementary Note 1, and the summary of phenotypic features are given in Table 1. Patients carrying either the p.(Arg116Leu) or the p.(Pro369Leu) mutation in *KCNQ1* displayed a wide endocrine phenotypic spectrum that ranged from relatively mild (only low IGF-1 levels in patients #10 and #11, or GF in subject #14) through classic GHD to multiple pituitary hormone deficiencies (Supplementary Note 1, Table 1). Several mutation carriers had GF (Fig. 1b) and mild craniofacial dysmorphic features, but only when the mutation was inherited from the mother (Table 1). One patient had retinal pigmentation (Supplementary Fig. 1). We therefore verified the expression of KCNQ1 in the developing human retina (Fig. 1b).

*KCNQ1* mutations are associated with alterations in the QT interval duration. Five of the 12 mutation carriers examined

(42%, 95% confidence interval 20–68%) displayed a corrected QT interval below the 2nd percentile of age- and gender-matched reference values, and two of them fulfilled the diagnostic criteria for short QT syndrome (Table 1)[10–12].

**Molecular modeling of KCNQ1**. We mapped the location of the two mutations to the 3D channel structure (Fig. 1c). KCNQ1 has six transmembrane segments (S1–S6) and intracellular N- and C-termini (CT). Arg116 is located in the N-terminus and Pro369 at the C-terminus. Both residues are proximal to the membrane[13]. Arg116 interacts with and Pro369 maps to an intracellular gating module, comprised of the proximal CT and constitutively bound CaM. The gating module directly connects to and regulates the activation gate in the sixth transmembrane segment (S6) that is responsible for opening and closing the channel pore[14] (Fig. 1c).

Functional channel expression requires calmodulin (CaM), the ubiquitous calcium binding protein[15,16] that serves as an obligate subunit of the channel complex, bound intracellularly. The biochemical properties of the Pro369Leu mutant was tested by using a pull-down assay of the intracellular CT to test binding of CaM. However, the Pro369Leu mutation does not compromise the structural integrity of the proximal C-terminal (CT)/CaM complex (Supplementary Figs. 2 and 3).

**Patch-clamp analyses and heart phenotypes of patients**. The electrophysiological properties of the mutated channels were examined in whole-cell patch-clamp analyses in HEK 293 cells, in which both mutated channels (p.Arg116Leu and p.Pro369Leu)

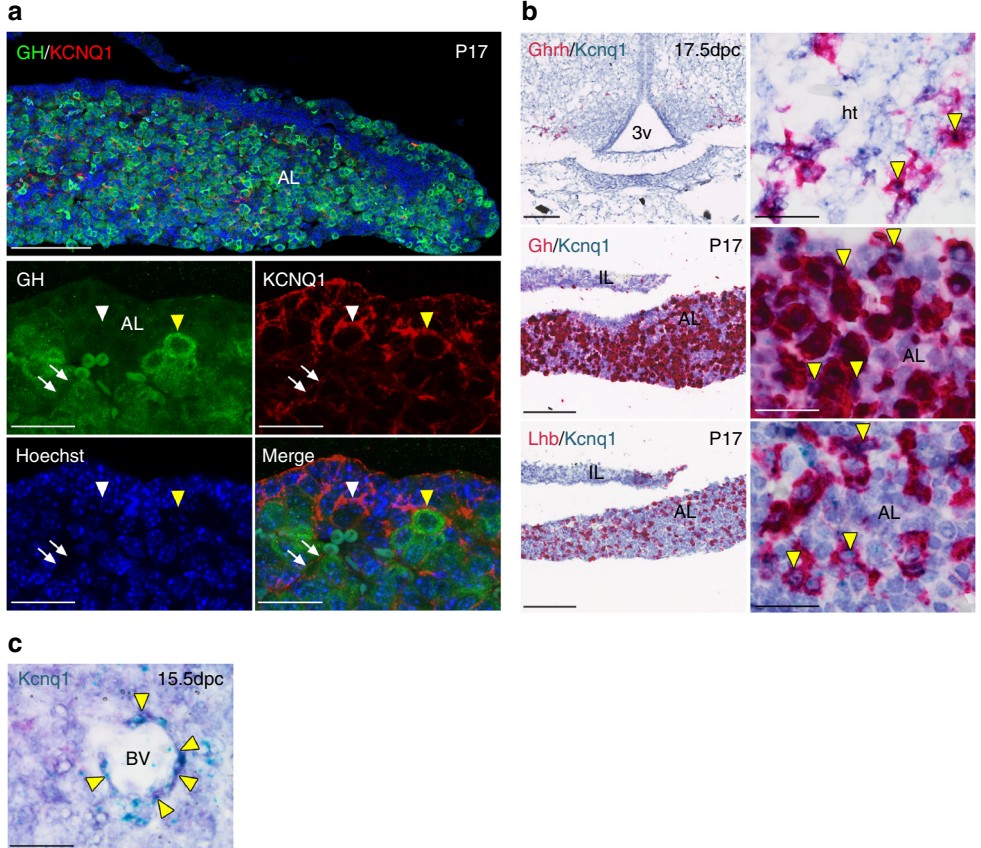

**Fig. 3** KCNQ1 is expressed in cells of the mouse hypothalamic–pituitary growth hormone axis. **a** Immunofluorescence staining against KCNQ1 (red) and Growth Hormone (GH, green) reveals membranous expression of KCNQ1 in a subset of GH-expressing cells of the postnatal pituitary at day 17 (P17, yellow arrowhead) as well as expression in additional pituitary cell types (white arrowhead) and surrounding blood vessels (arrows). **b** Sensitive RNAscope in situ hybridization detects *Kcnq1* mRNA (aqua) in *Ghrh*-expressing neurons of the developing hypothalamus (red) at 17.5 days post coitum (dpc), and confirms the expression in *Gh*-expressing somatotropes (red) at P17. *Kcnq1* is expressed in additional endocrine cells types such as *Lhb*-expressing gonadotropes (red). Examples of double-positive cells are noted by yellow arrowheads. **c** *Kcnq1* mRNA is detected in cells surrounding blood vessels of the developing anterior pituitary at 15.5 dpc. Scalebars indicate 100 μm in low magnification images and 20 μm in high magnification images. *AL* anterior lobe, *BV* blood vessel, *Ht* hypothalamus, *IL* intermediate lobe, *3v* third ventricle

gave higher current levels than the wild-type (WT) Kv7.1 channels when expressed alone ($P < 0.05$–$0.001$), and notably, the activation kinetics of KCNQ1-Arg116Leu was shifted to more depolarized values (Fig. 2). Potassium channels comprised of KCNQ1 and KCNE1 subunits are responsible for one of the primary cardiac repolarizing currents, the slow delayed rectifier current ($I_{Ks}$)[17], and mutations in *KCNQ1* are known to cause the long QT syndrome[7,8] (LQT1, MIM: 192500) and the rare short QT syndrome[8,9] (MIM: 609621). The electrophysiological properties of the mutated channels were therefore also examined when KCNQ1 was co-expressed with KCNE1. Ectopically expressed KCNQ1-Pro369Leu/KCNE1 Kv7.1 channels gave profoundly increased currents (Fig. 2), whereas the currents produced by KCNQ1-Arg116Leu/KCNE1 Kv7.1 channels at +60 mV did not differ from the control, but still, the voltage dependence of activation kinetics were shifted to more depolarized potentials (Fig. 2). Co-expressing KCNQ1 with KCNE2, another β subunit expressed in a number of other tissues together with KCNQ1[18,19], showed that both KCNQ1-Pro369Leu/KCNE2 and KCNQ1-Arg116Leu/KCNE2 increased the current levels ($P < 0.05$–$0.001$) (Fig. 2). Taken together, these results suggests that the Pro369Leu and Arg116Leu mutations impart subtle but distinct conformational changes to this module that favor the open channel state, in particular when assembled with KCNE2.

**Expression of KCNQ1 in mouse and human**. We studied the expression of KCNQ1 in growth-regulating hypothalamic and pituitary cell types in mice by using immunofluorescence and mRNA in situ hybridization techniques. *KCNQ1* expression was also examined by RT-PCR in human hypothalamic and pituitary gland cDNA. Our results show that KCNQ1 is expressed in mouse somatotrope and gonadotrope cells in the postnatal pituitary (Fig. 3a, b), and in the human pituitary (Supplementary Fig. 4a). *Kcnq1* was also expressed in mouse hypothalamic GHRH neurons during development, and in the human hypothalamus (Fig. 3b and Supplementary Fig. 4b). Finally, Kcnq1 was expressed along the blood vessels of the developing mouse pituitary, and KCNQ1was detected along the postnatal pituitary capillaries by immunostaining (Fig. 3a, c). We also verified the expression of *KCNE2* in human hypothalamus and pituitary (Supplementary Fig. 4c).

**ACTH secretion assay and protein expression quantification.** The impact of KCNQ1 on pituitary hormone secretion was investigated by transiently overexpressing WT or one of the mutant KCNQ1s (p.Arg116Leu, p.Pro369Leu) in the well-characterized, ACTH-secreting mouse pituitary tumor cell line AtT-20/D16v-F2, widely used to investigate the regulation of exocytosis by voltage-gated calcium entry[20,21]. The LQT1-causing

**Table 2 Mixed linear regression model estimates for ACTH concentrations (ng/ml) measured in diluted medium samples from cells transfected with KCNQ1**

| Parameter | Estimate | SE | p |
|---|---|---|---|
| Intercept (WT KCNQ1)[a] | 7.26 | 0.99 | <0.001 |
| Time | 0.76 | 0.25 | 0.004 |
| KCNQ1-Arg116Leu | 0.32 | 0.72 | 0.659 |
| KCNQ1-Pro369Leu | 0.62 | 0.72 | 0.393 |
| KCNQ1-Gly589Asp | 1.11 | 0.72 | 0.126 |

Mixed linear regression model estimates for ACTH concentrations (ng/ml) measured in diluted medium samples (1:10) in the reference condition (WT KCNQ1) and three other KCNQ1 conditions (KCNQ1-Arg116Leu, KCNQ1-Pro369Leu, KCNQ1-Gly589Asp), their standard errors (SE), and corresponding P values. The model included both the growth time (per 1 h) and the KCNQ1 conditions as fixed effects and the experiment number as a random intercept. ACTH levels produced by the cells transfected with mutant KCNQ1s did not differ from the levels produced by the WT KCNQ1. Similarly, the CRF-induced ACTH secretion did not differ between the four KCNQ1 environments (data not shown)
[a]At −2 h

p.Gly589Asp mutant, which is a founder mutation in the Finnish population, served as an additional control. ACTH levels produced by cells transfected with the mutant KCNQ1s did not differ from the reference condition (WT KCNQ1) (Table 2). In contrast, ACTH levels produced by cells transfected with KCNQ1-Arg116Leu or KCNQ1-Pro369Leu together with KCNE2 were significantly lower than those produced by the cells transfected with the WT KCNQ1 and KCNE2 ($P < 0.001$ and $P = 0.004$, respectively) (Table 3). We subsequently examined whether the observed differences in ACTH levels could be explained by relative expression levels of the mutant constructs. Relative expression of the p.Pro369Leu did not differ from the WT KCNQ1, whereas the expression of p.Arg116Leu was lower than that of the WT KCNQ1 ($P < 0.05$) (Supplementary Figs. 5 and 6). The lowest relative expression was exhibited by the additional control (p.Gly589Asp) ($P < 0.005$ vs. WT KCNQ1). Importantly, however, the ACTH levels produced by the cells transfected with the KCNQ1-Gly589Asp and KCNE2 did not differ from the WT condition (Table 3), suggesting that the differences in relative KCNQ1 expression levels did not determine the ACTH levels measured in the culture medium.

**Protein–protein interaction studies**. We next examined the possibility that the two KCNQ1 mutations may disrupt critical protein–protein interactions. We explored this by expressing KCNQ1 in Flp-In T-REx 293 cells, and found 68 high-confidence interacting partners that were involved in processes such as spliceosomal snRNP assembly and intracellular trafficking (Supplementary Fig. 7a). However, the protein–protein interaction heatmap did not reveal gross differences among the strongest interactors between the WT KCNQ1 and KCNQ1-Arg116Leu or KCNQ1-Pro369Leu (Supplementary Fig. 7b).

## Discussion

We identified two missense mutations in KCNQ1 to underlie childhood onset of GHD and maternally inherited gingival fibromatosis. Very recently, maternally inherited variation in the KCNQ1 locus was shown to associate with reduction in adult height in the Sardinian population[22]. Inheritance of GHD in our families, however, followed an autosomal dominant pattern, and similarly, a lack of a parent-of-origin effect has been described in the long QT syndrome caused by loss-of-function KCNQ1 mutations[23]. The KCNQ1 locus resides in the imprinted region of chromosome 11p15, and during early ontogenesis, KCNQ1 is paternally imprinted. In our patients, only those with a maternally inherited KCNQ1 mutation displayed congenital GF, and several of them also had mild craniofacial dysmorphic features

that were mitigated by adulthood, suggesting loosening of KCNQ1 imprinting in cranial tissues with age, similarly as reported to occur in the pancreas[24]. The two twin pairs (monozygotic twins #7 and #8 and dizygotic twins #21 and #22), present in our families, are prismatic for understanding the phenotypic variability: subject #21 was smaller than his brother at birth, and had a more severe endocrine phenotype (Supplementary Note 1, Table 1). Similarly, subject #7, who was born breech and smaller than his monozygotic twin brother, had more severe growth failure during childhood (Supplementary Note 1, Table 1). Thus, environmental factors are likely to modify the endocrine phenotype of this syndrome. We also found a spectrum of corrected QT intervals in the mutation carriers, which indicates incomplete penetrance and variable expressivity of the mutations in the heart, similarly as described for long QT syndrome due to loss-of-function KCNQ1 mutations[25].

Our results show that KCNQ1 is expressed in mouse somatotrope and gonadotrope cells in the postnatal pituitary and in the human pituitary. Indeed, previous findings have implicated voltage-gated potassium channel currents in pituitary cells in different species[26–28]. Kcnq1 was also expressed in mouse hypothalamic GHRH neurons during development, and in the human hypothalamus. These findings raise the possibility that the Arg116Leu and Pro369Leu KCNQ1 mutations may impact hypothalamic–pituitary function at multiple levels; in the case of the growth hormone axis, both on the control of episodic GHRH secretion by hypothalamic neurons as well as on somatotrope function during growth. Kcnq1 was also expressed along the blood vessels of the developing mouse pituitary, and KCNQ1 was detected along the postnatal pituitary capillaries by immunostaining. Somatotropes are organized in the close vicinity of pituitary blood vessels[29], and GHRH enhances the oxygen supply to the somatotrope network via increased capillary blood flow[29,30]. The anterior pituitary is richly vascularized by fenestrated capillaries emanating from the pituitary portal system, and direct arterial supply has also been demonstrated in human pituitary adenomas[31–33]. Although voltage-gated potassium channels, including KCNQ1, are implicated in the vascular physiology[34], their possible role, if any, in the regulation of pituitary blood flow is currently unknown.

Although the exact mechanism by which the two KCNQ1 mutations cause pituitary hormone deficiency in humans is unclear, our data suggest that the KCNQ1–KCNE2 complex may play a role in it. Co-expressing KCNQ1 with KCNE2 β-subunit showed that both KCNQ1 mutants increased current levels in patch clamp analyses and were associated with reduced pituitary hormone secretion from AtT-20 cells. Both KCNQ1 and KCNE2 have been implicated in exocytosis[35,36], and our electrophysiological and hormone secretion data suggest that the p.Arg116Leu and p.Pro369Leu mutations increase potassium conductance, which via hyperpolarization leads to diminished hormone secretion. The presence of KCNE2 expression in human hypothalamus and pituitary tissues is in agreement with the previously observed expression of Kcne2 in rodents[37,38], and thus lends credence to the above hypothesis. Intriguingly, KCNE2 can regulate several different Kv channels, and plays an important role in the endocrine system. Kcne2 deletion causes hypothyroidism in pregnant and lactating mice and in their pups[18], and very recently $Kcne2^{-/-}$ mice were reported to have impaired glucose-induced insulin secretion[36]. In light of the mutations' location in the 3D channel structure, the electrophysiological findings, and our observation that neither mutation compromises the integrity of the proximal CT/CaM gating module, i.e., CaM association with the CT, we conclude that the Pro369Leu and Arg116Leu mutations probably favor the open channel state especially when assembled with the KCNE2 β subunit.

**Table 3 Mixed linear regression model estimates for ACTH concentrations (ng/ml) measured in diluted medium samples from cells transfected with *KCNQ1* and *KCNE2***

| Parameter | Estimate | SE | P |
|---|---|---|---|
| Intercept (WT KCNQ1/KCNE2)[a] | 9.59 | 0.84 | <0.001 |
| Time | 0.62 | 0.15 | <0.001 |
| KCNQ1-Arg116Leu/KCNE2 | −1.88 | 0.42 | <0.001 |
| KCNQ1-Pro369Leu/KCNE2 | −1.29 | 0.42 | 0.004 |
| KCNQ1-Gly589Asp/KCNE2 | −0.77 | 0.42 | 0.075 |

Mixed linear regression model estimates for ACTH concentrations (ng/ml) measured in diluted medium samples (1:10) in the reference condition (WT KCNQ1/KCNE2) and three other KCNQ1 conditions (KCNQ1-Arg116Leu/KCNE2, KCNQ1-Pro369Leu/KCNE2, KCNQ1-Gly589Asp/KCNE2), their standard errors (SE), and corresponding P values. The model included both the growth time (per 1 h) and the KCNQ1/KCNE2 conditions as fixed effects and the experiment number as a random intercept. Based on the regression model, the KCNQ1-Arg116Leu with KCNE2 produced 20% (−2 h) or 17% (0 h) less ACTH than the reference (WT KCNQ1 with KCNE2) (P < 0.001). Similarly, the KCNQ1-Pro369Leu with KCNE2 produced approximately −12–13% less ACTH (P = 0.004) than the reference environment. ACTH levels produced by the cells transfected with the LQT1 mutant KCNQ1-Gly589Asp/KCNE2, which served as a control, did not differ significantly from the reference environment. The CRF-induced ACTH responses between the four KCNQ1/KCNE2 conditions did not differ (data not shown)
[a]At −2 h

The protein–protein interaction heatmap did not reveal gross differences among the strongest interactors between the WT KCNQ1 and KCNQ1-Arg116Leu or KCNQ1-Pro369Leu. Thus, rather than resulting from a specific protein–protein interaction defect(s), this syndrome may primarily arise from altered potassium channel balance in different cell types, with variable manifestations depending on the interplay with environmental factors. Indeed, patients that carry mutations in other potassium channels share phenotypic similarities with our patients: a craniofacial phenotype, GHD and hypogonadotropic hypogonadism have been reported in a patient with Cantù syndrome (MIM: 239850)[39], caused by gain-of-function mutations in *ABCC9* or *KCNJ8*[40], gingival fibromatosis/overgrowth in patients carrying mutations in *KCNJ6*[41] and *KCNH1*[42], and a craniofacial phenotype in patients with mutations in *KCNJ6*[41], *KCNJ2*[43], and *KCNK9*[44]. One possible explanation for the cranial phenotype and GF stems from altered potassium channel balance in the cranial neural crest[45], a transient cell population which gives rise to the bone and cartilage in the cranial area, and to 90% of gingival mesenchymal stem cells[46]. Of note, cyclosporine A has been proposed to activate KCNQ1[47], and this immunosuppressive agent also frequently causes GF as a side effect. Thus, it is possible that KCNQ1 activation plays also a role in drug-induced GF.

In conclusion, our results reveal a role for the potassium channel protein KCNQ1 in the regulation of human growth, and show that GHD associated with maternally inherited gingival fibromatosis, long QT syndrome[7,8], short QT syndrome[8,9], familial atrial fibrillation[48] (MIM: 607554), and Jervell and Lange-Nielsen cardioauditory syndrome[49] (MIM: 220400) are allelic disorders. Ion channels are known to regulate several processes in the pituitary cells[26,50]. Our work builds on these previous observations and demonstrates that ion channels are clinically relevant regulators of pituitary function in humans.

## Methods

**Ethics.** This study was approved by the ethics committees of the Oulu University Hospital, Finland, and the Helsinki University Central Hospital, Finland. The DNA from family members recruited in Switzerland was obtained by permission of the Swiss Society of Medical Genetics. All participants gave written informed consent.

**Subjects.** Members of three unrelated families were recruited: a large Finnish family, a second Finnish family, and a Swiss family originating from Argentina. Their phenotypes were ascertained by physical examination, medical records, or, occasionally, self-report. Phenotypic information on patients 5–9 (Fig. 1) was described earlier[4]. DNA was extracted from peripheral blood leukocytes of the subjects (once from saliva, DNA Genotek, Inc.). A 12-lead ECG was obtained from 12 mutation carriers for evaluation of conduction times. Corrected QT interval (QTc) was calculated from the QT and inter-beat (RR) interval using Bazett's correction (QTc=QT/(RR)$^{1/2}$). Hormone measurements were done in the clinical laboratory.

**Linkage analysis.** We performed genome-wide single-nucleotide polymorphism (SNP) genotyping of 16 family members (Illumina HumanOmniExpress microarray, Institute for Molecular Medicine Finland) of the large Finnish pedigree. Two-point parametric linkage analysis was performed using Pseudomarker 1.0.6 software with FASTLINK 4.1P linkage package[51,52]. Individuals with GF and/or GHD with short stature or low insulin-like growth factor 1 (IGF-1) levels were treated as affected; all others were considered as affected. A dominant model of inheritance with nearly complete penetrance was assumed: disease allele frequency was set to 0.0001 with a penetrance rate of 0.999 and a phenocopy rate of 0.001. Allele frequencies were calculated from the genotypes of all individuals.

**Identification of shared variants.** Shared variants within the linkage region in two affected members of the large Finnish family (Fig. 1a) were examined by whole genome sequencing (Beijing Genomics Institute). Novel variants in the linkage region shared between the two genomes and not found in any databases (dbSNP, http://www.ncbi.nlm.nih.gov/SNP/; Exome Variant Server, http://evs.gs.washington.edu/EVS/; 1000 Genomes, http://www.1000genomes.org/) were annotated with SNPnexus (http://www.snp-nexus.org/).

**Segregation analyses and sanger sequencing of *KCNQ1*.** The Arg116Leu missense mutation in *KCNQ1* was Sanger-sequenced in all 16 family members and in four additional affected family members participating in the study after completion of linkage analysis. The mutation was also screened in 200 healthy controls (anonymous blood donors from the Finnish Red Cross Blood Service). Additional rare variants shared between the two genomes, upstream and downstream of *KCNQ1*, were genotyped in the family to construct haplotypes. Coding exons and exon–intron boundaries of *KCNQ1* were then sequenced in the second Finnish family (mother and two boys), as well as in the index patient from the family originating from Argentina and her parents (Fig. 1a). The primers used in the Sanger sequencing of *KCNQ1* are listed in Supplementary Table 1. PCR conditions are available upon request.

**Electrophysiological studies.** The electrophysiological properties of the Arg116Leu and Pro369Leu KCNQ1 mutants were studied by whole-cell patch clamping, as described[53]. In brief, the two mutations were introduced into the pXOOM-hKCNQ1 expression plasmid. HEK293 cells were transiently cotransfected with 1 μg pXOOM-hKCNQ1 (WT or mutants), 1 μg pcDNA3-hKCNE1/hKCNE2 for the KCNQ1/KCNE experiments, and 0.2 μg of pcDNA3-eGFP as a reporter gene, using Lipofectamine 2000 (Invitrogen), according to the manufacturer's instructions. Patch-clamp experiments were performed at room temperature (20–22 °C) 2–3 days after transfection. Patch-clamp recordings were conducted using an internal solution containing the following (mM): KCl (140), Na$_2$-ATP (1), EGTA (2), HEPES (10), CaCl$_2$ (0.1), MgCl$_2$ (1), and D-glucose (10), pH 7.4 with KOH; external solution NaCl (140), KCl (4), CaCl$_2$ (2), MgCl$_2$ (1), HEPES (10), and D-glucose (10), pH 7.3 with NaOH. Cells expressing WT or mutant KCNQ1 potassium channels, detected by green fluorescence, were subjected to voltage clamping to detect potential changes in activation and inactivation kinetics.

**Statistics for electrophysiological studies.** For electrophysiological studies, mean ± SEM values are shown. Statistical significance was evaluated by two-way analysis of variance (ANOVA) with Bonferroni correction. P < 0.05 was considered statistically significant.

**ACTH secretion studies and western blotting.** AtT20-20/D16v-F2 cells (Sigma) were maintained in DMEM (Sigma) supplemented with 10% FBS (v/v) (HyClone), penicillin (25 U/ml, Sigma), streptomycin (25 U/ml, Sigma), and 2 mM glutamine (Invitrogen). At the beginning of each experiment, the cells were seeded on 24-well

plates at a density of 90,000 cells per well. The plates were incubated overnight at 37 °C in a humidified atmosphere of 5% $CO_2$–95% air, and transfected using Lipofectamine 3000 transfection reagent (Invitrogen) according to manufacturer's instructions. Each well received either 500 ng of pXOOM-hKCNQ1 (WT or mutants Arg116Leu, Pro369Leu, Gly589Asp) alone, or 250 ng of pXOOM-hKCNQ1 (WT or mutants) together with 250 ng of pcDNA3-hKCNE2 (two wells for each condition). Thirty hours after transfection, culture medium was changed to DMEM containing 0.2% (vol/vol) FBS to starve the cells. The first medium samples for ACTH measurements were collected ~16 h later (−2 h sample) and the second samples 2 h thereafter (0 h sample). After that, 100 nM CRH (Bachem) was added to each well, and the third sample was collected 2 h later (+2 h sample). Each sample was collected by combining media from duplicate wells and samples were stored at −80 °C until ACTH concentrations were analyzed in diluted aliquots (1:10) with a commercially available ELISA kit (Phoenix pharmaceuticals, Inc.) with a sensitivity of 0.08 ng/ml, according to the instructions of the manufacturer. This experiment was repeated seven times. The data were analyzed as explained the section Statistical analysis of ACTH secretion studies.

For Western blotting, cells transfected with 250 ng of pXOOM-hKCNQ1 (WT or mutants) together with 250 ng of pcDNA3-hKCNE2 were collected and combined from duplicate wells 48 h after transfection and lysed in 2× Laemmli buffer and an equal volume of 8 M urea was added. Lysates were sonicated, incubated at 37 °C for 30 min, and loaded on a 4–20% gradient gel (Bio-Rad). The proteins were transferred onto a nitrocellulose membrane and detected with 1:1000 rabbit polyclonal α-KCNQ1 antibody (H-130, sc-20816, Santa-Cruz Biotechnology) and 1:3000 goat α-rabbit IgG (H+L)-HRP conjugate secondary antibody (#1706515, Bio-Rad) (Supplementary Fig. 5). To control for total protein loaded on the gel, the blot was stripped with Restore Western Blot Stripping Buffer (Thermo Scientific) and β-actin was detected with 1:2000 mouse monoclonal α-β-actin antibody (C4, sc-47778, Santa-Cruz Biotechnology) and 1:3000 goat α-mouse IgG (H+L)-HRP conjugate secondary antibody (#1706516, Bio-Rad). KCNQ1 expression was quantitated from three independent replicates by using Image Studio Lite software (LI-COR Biotechnology). Briefly, the relative density of all mutants in relation to WT KCNQ1 was calculated, and the values were adjusted by using the relative density of β-actin of the corresponding lane.

**Statistical analysis of ACTH secretion studies.** Two different linear mixed models were used to explain the ACTH levels measured in diluted (1:10) cell culture media samples that were obtained after 16 h (−2 h) and 18 h (0 h) of serum starvation. In the first model, the cells had been transfected with four different KCNQ1s alone (WT KCNQ1, KCNQ1-Arg116Leu, KCNQ1-Pro369Leu, or KCNQ1-Gly589Asp), and, in the second model, the cells had been transfected with one of the four KCNQ1 constructs along with the construct encoding the beta subunit KCNE2.

In both models, the experiment number was included as a random intercept, and the KCNQ1 environment (WT KCNQ1, KCNQ1-Arg116Leu, KCNQ1-Pro369Leu, or KCNQ1-Gly589Asp), and time (−2 h, 0 h) were included as fixed effects. In both models, the WT KCNQ1 was used as the reference KCNQ1 environment. The residual plots were inspected for deviations from normality and homoscedasticity, and the $P$-values were calculated using Satterthwaite's approximation for the degrees of freedom. These analyses were carried out using R (version 3.3.3) with packages lme4[54] (version 1.1-13) (https://cran.r-project.org/package=lme4) and lmerTest (version 2.0-33) (https://CRAN.R-project.org/package=lmerTest).

The CRF-induced increases in ACTH secretion were estimated by calculating fold inductions for each four environments by dividing the ACTH concentrations, measured 2 h after CRF administration (+2 h), by the baseline concentration (average of ACTH concentrations measured at −2 h and 0 h), and subjecting these values to ANOVA. These analyses were carried out using Graph Pad Prism (version 7.01). $P < 0.05$ was accepted to indicate statistical significance.

**Immunofluorescence and mRNA in situ hybridization.** Wild-type CD1 mouse embryos and postnatal pituitaries were fixed in 4% neutral buffered formalin for 16 h, dehydrated through an increasing ethanol series and paraffin-embedded using standard protocols. Sections were cut at 4 µm and mounted on Superfrost plus slides. For immunofluorescence, antigen retrieval was carried out in a pressure cooker in citrate buffer, pH 6.0. Fluorescence detection was performed using specific antibodies against KCNQ1 (1:300 dilution, Santa Cruz sc365186 mouse monoclonal) and against GH (1:1000 dilution, National Hormone and Peptide Program AFP-564180 rabbit polyclonal, detected with goat anti-rabbit 488 1:300 dilution, Thermo Fisher A11008), with biotin amplification for KCNQ1 (biotinylated goat anti-mouse 1:350 dilution, Abcam ab6788; Streptavidin-555 1:500, Thermo Fisher S32355). Slides were incubated with Hoechst (1:10,000, Thermo Fisher H3570) before mounting. For duplex mRNA in situ hybridization, slides were pretreated by boiling for 10 min[55] and hybridized with specific probes against *Kcnq1* (channel 1, Cat. No. 507611 target region 827-2024—exons 5–16 (NM_008434.2)), *Gh* (channel 2, Cat. No. 445361-C2 target region 9-630—exons 1–5 (NM_008117.3)), *Ghrh* (channel 2, Cat. No. 470991-C2 target region 2-483—exons 1–5 (NM_010285.3)) and *Lhb* (channel 2, Cat. No. 478401-C2 target region

26-503—exons 2–3 (NM_008497.2)) (all ACDBio). The RNAscope 2.5 HD Duplex Assay kit (ACDBio) was used according to the manufacturer's recommendations, with the following adaptations to increase signal detection: incubation in amplification 9 was extended to 1 h, and the Green B to Green A reagent ratio was increased to 1:30.

**RT-PCR amplification of KCNQ1.** A 213-bp fragment of *KCNQ1* complementary DNA (cDNA) was amplified from human hypothalamic cDNA (Clontech, prepared from the hypothalami of 30 male/female Caucasians, aged 15–68 years, after sudden death), or, as a positive control, from the cardiac cDNA library of a 59-year-old male donor (Amsbio) with cDNA-specific primers (Supplementary Table 1, HuKCNQ1_F+R). β-actin served as a reference gene. Similarly, a 186-bp fragment of *KCNQ1* cDNA was PCR-amplified from human pituitary gland cDNA (Clontech, normal pituitary gland pooled from 12 male/female Caucasians, ages 18–52; cause of death: trauma) with cDNA specific primers (Supplementary Table 1, HuKCNQ1_F2+R2). *GAPDH* served as a reference gene. Additionally, a 172-bp fragment of *KCNE2* cDNA was amplified from the human pituitary gland cDNA and the human hypothalamic cDNA with cDNA-specific primers. The PCR products were visualized on 1.5% agarose gel (Supplementary Fig. 4).

**Cloning of the KCNQ1 expression vectors for AP-MS.** To generate Gateway entry clones from Human *KCNQ1* and the Arg116Leu, Pro369Leu, Gly589Asp mutants, flanking attB sites were generated using PCR, followed by Gateway BP reaction. The corresponding entry clones were then used to generate C-terminally tagged expression vectors (pcDNA5/FRT/TO/BioID/StrepII/HA/GW) via Gateway LR recombination.

**Affinity purification.** For affinity purification, tetracycline-inducible Flp-In T-REx 293 cell lines expressing KCNQ1 wild type and other mutations were generated as described[56]. In brief, the cells were co-transfected with the tagged-*KCNQ1* expression constructs and pOG44 vector (for Flp-recombinase expression, Invitrogen) using FuGENE 6 transfection reagent (Promega). After 2 days, transfected cells were selected in Hygromycin B (100 mg/ml; Invitrogen) containing medium for 3 weeks. The positive clones containing stable isogenic tagged-KCNQ1 (wt and mutants) were expanded and for each construct ~$5 \times 10^7$ cells ($5 \times 15$ cm dishes) in three biological replicates were induced with 1 µg/ml doxycycline (MP Biomedicals) for 24 h. After induction, cells were washed with 0.1 mM $MgCl_2$, 0.1 mM $CaCl_2$ in PBS and harvested in 1 mM EDTA-PBS. Cells were pelleted by centrifugation $400 \times g$, 5 min, 4 °C. Cell pellets (~$5 \times 10^7$ cells) were lysed in 3 ml of HENN-lysis buffer (50 mM HEPES pH 8.0, 5 mM EDTA, 150 mM NaCl, 50 mM NaF, 0.5% NP40, 1 mM DTT, 1.5 mM $Na_3VO_4$, 1 mM PMSF, and 1× protease inhibitors cocktail; Sigma) on ice. Cleared cell lysates were loaded on spin columns (Bio-Rad, USA) containing 200 ml of Strep-Tactin beads (IBA GmbH), washed thrice with 1 ml HENN-lysis buffer and 4× with 1 ml HENN buffer. Bound proteins were eluted with 900 µl of elution buffer (0.5 mM Biotin; Pierce).

**Mass spectrometry.** For liquid chromatography–mass spectrometry (LC–MS/MS) samples were prepared as follows: cysteine bonds were reduced with 5 mM Tris (2-carboxyethyl)phosphine (TCEP) (Sigma-Aldrich) for 20 min at 37 °C and alkylated with 10 mM iodoacetamide (Fluka, Sigma-Aldrich) for 20 min at room temperature in the dark. A total of 1 µg trypsin (Promega) was added and samples digested overnight at 37 °C. Samples were quenched with 10% trifluoroacetic acid (TFA) and purified with C-18 Micro SpinColumns (The Nest Group) eluting the samples to 0.1% TFA in 50% acetonitrile (ACN). Samples were dried by vacuum concentration and peptides were reconstituted in 30 µl buffer A (0.1% TFA and 1% ACN in LC–MS grade water) and vortexed thoroughly.

LC–MS/MS analysis was performed on an Orbitrap Elite ETD hybrid mass spectrometer using the Xcalibur version 2.2 SP 1.48 coupled to EASY-nLCII-system (all from Thermo Fisher Scientific) via a nanoelectrospray ion source. In total, 6 µl and 5 µl of peptides were loaded from Strep-Tag and BioID-samples, respectively. Samples were separated using a two-column setup consisting of a C18 trap column (EASY-Column 2 cm × 100 µm, 5 µm, 120 Å, Thermo Fisher Scientific), followed by C18 analytical column (EASY-Column 10 cm × 75 µm, 3 µm, 120 Å, Thermo Fisher Scientific). Peptides were eluted from the analytical column with a 60 min linear gradient from 5 to 35% buffer B (buffer A: 0.1% FA, 0.01% TFA in 1% acetonitrile; buffer B: 0.1% FA, 0.01% TFA in 98% acetonitrile). This was followed by 5 min 80% buffer B, 1 min 100% buffer B followed by 9 min column wash with 100% buffer B at a constant flow rate of 300 nl/min. Analysis was performed in data-dependent acquisition mode where a high resolution (60,000) FTMS full scan ($m/z$ 300–1700) was followed by top20 CID-MS2 scans (energy 35) in ion trap. Maximum fill time allowed for the FTMS was 200 ms (Full AGC target 1,000,000) and 200 ms for the ion trap (MSn AGC target 50,000). Precursor ions with more than 500 ion counts were allowed for MSn. To enable the high resolution in FTMS scan preview mode was used.

**Data processing.** SEQUEST search algorithm in Proteome Discoverer software (Thermo Fisher Scientific) was used for peak extraction and protein identification with the human reference proteome of UniProtKB/SwissProt database (www.uniprot.org). Allowed error tolerances were 15 ppm and 0.8 Da for the precursor

and fragment ions, respectively. Database searches were limited to fully tryptic peptides allowing one missed cleavage (in peptide mapping semi-tryptic with one missed cleavage allowed), and carbamidomethyl +57,021 Da (C) of cysteine residue was set as fixed, and oxidation of methionine +15,995 Da (M) as dynamic modifications. For peptide identification FDR was set to <0.05.

The high-confidence protein–protein interactions were identified using stringent filtering against control contaminant database. The bait normalized relative protein abundances (% to the bait) were calculated from the spectral counts. Each average and SD was calculated from three biological replicates. The high confidence protein–protein interactions data were imported into Cytoscape 3.4.0 for the visualization combined with the known protein-protein interactions from PINA2 (Protein Interaction Network Analysis http://cbg.garvan.unsw.edu.au/pina/)[57]. Protein complexes were annotated against CORUM database[58] and Gene Ontology term of biological pathway enrichment analysis was using DAVID (https://david.ncifcrf.gov/)[59] (Supplementary Fig. 7).

**Molecular modeling of the KCNQ1 Pro369Leu mutant.** Coordinates for the KCNQ1 proximal CT domain in complex with CaM (PDB code 4V0C), derived from the crystal structure[14], were used to prepare the model. Pro369 was mutated to Leu, selecting the most populated rotamer using Coot[60]. The molecular graphics were prepared in PyMol (Supplementary Fig. 2a).

**Pull-down assay.** The three KCNQ1 Pro369Leu mutant CT constructs (pET Duet (Merck) with sequential HisTag and TEV protease sequences fused to (i) KCNQ1 352-622 H620A Δ406–504; (ii) 352–539; (iii) 352–539 Δ406–504 in multi-cloning site I with CaM inserted into multi-cloning site II) were co-expressed with CaM as described[14]. Cell paste was suspended in lysis buffer (50 mM sodium phosphate pH = 8, 300 mM NaCl, 0.1% Triton X-100), sonicated, and clarified by centrifugation. Supernatant (2 ml) was incubated with $Ni^{2+}$-NTA beads (50 μl) for 1 h at 4 °C. Beads were then washed three times with wash buffer (50 mM sodium phosphate pH = 8, 300 mM NaCl, 0.5% Triton X-100, 15 mM imidazole). A sample of the final wash was kept for analysis. Elution from beads was performed by incubation for 15 min with elution buffer (50 mM sodium phosphate pH = 8, 300 mM NaCl, 250 mM imidazole). Samples were analyzed by SDS-PAGE and immunoblot was probed with anti-6×His-peroxidase mouse monoclonal (Roche, cat. no. 11965085001, 1:500 dilution of manufacturer's suspended lyophilizate) and anti-CaM mouse monoclonal (Millipore, cat. no. 05-173, 1:1000 dilution of supplied manufacturer's solution) antibodies (Supplementary Fig. 2b).

**Immunohistochemistry in human samples.** Human fetuses (11 weeks post amenorrhea, $n = 3$) were immersion-fixed in 4% paraformaldehyde in 0.1 M phosphate-buffer saline (PBS) (pH 7.4) for 3 weeks at 4 °C, cryoprotected in 30% sucrose in PBS for 48 h, embedded in Tissue Tek (Miles), and frozen in liquid nitrogen. Tissues were cryosectioned at 16 μm. Localization of KCNQ1 was accomplished by using immunofluorescence procedures as previously described[61] (Supplementary Fig. 1). Briefly, sections were blocked in an incubation solution containing PBS pH 7.4, 0.3% Triton X-100 (TBS-T; Sigma, T8787) with 10% normal donkey serum (NDS; D9663; Sigma) for 2-h at room temperature (RT). After blocking, sections were incubated for 48-h at 4 °C with a rabbit polyclonal anti-KV7.1 (KCNQ1) antibody (#APC-022, Alomone labs) diluted 1:200 in a solution containing PBS pH 7.4, 0.3% Triton X-100 and 10% normal donkey serum. For controls in each immunolocalization procedure, PBS was substituted for the primary antibody. After PBS rinses, immunoreactivity was revealed using an Alexa-Fluor 488-conjugated secondary antibody (1:400; Life Technologies, reference Molecular Probes, Invitrogen, A21206) for 90-min at RT. After PBS washes, sections were incubated with 0.02% Hoechst (H3569; Invitrogen) in PBS for 15-min at RT and coverslipped with mowiol medium (Sigma, Cat. #81381).

Sections were examined using an Axio Imager.Z1 ApoTome microscope (Carl Zeiss, Germany) equipped with a motorized stage and an AxioCam MRm camera (Zeiss). For confocal observation and analyses, an inverted laser scanning Axio observer microscope (LSM 710, Zeiss) with an EC Plan NeoFluor×100/1.4 numerical aperture oil-immersion objective (Zeiss) was used with an argon laser exciting at 488 nm (Imaging Core Facility of IFR114, of the University of Lille 2, France).

**Subject details and ethics.** Human embryos (<GW9) and fetuses (>GW9) were obtained from voluntarily terminated pregnancies with the parent's written informed consent (Gynaecology Hospital Jeanne de Flandres, Lille, France). Tissues were made available in accordance with the French bylaw (Good practice concerning the conservation, transformation and transportation of human tissue to be used therapeutically, published on December 29, 1998).

Permission to use non-pathological human fetal tissues was obtained from the French Agency for Biomedical Research (Agence de la Biomédicine, Saint-Denis la Plaine, France, protocol no. PFS16-002).

**Data availability.** All relevant data are available from the authors.

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

## Acknowledgements

Ms. Lea Puhakka is thanked for skillful technical assistance. Dr. Päivi Miettinen is thanked for commenting on the manuscript. This work was supported by the Academy of Finland (138124 to J.T., 251413 to T.R., 294173 to M.V.), Foundation for Pediatric Research (7495 to T.R.), Sigrid Juselius Foundation (2613 to T.R.), Emil Aaltonen Foundation (2170 to T.R.), Novo Nordisk Foundation (4761 to T.R.), Helsinki University Central Hospital research funds (2010307), Jalmari and Rauha Ahokas Foundation (to J. T.), Paulo Foundation (to J.T.), Danish Council for Independent Research (DFF-1331-00313B to T.J.), Agence Nationale de la Recherche, ANR, France (ANR-14-CE12-0015-01 RoSes and GnRH to P.G.: ANR 12 BSV1 0032 Peri-Pulse to both P.G. and P.M.), Swiss National Science Foundation grants (31003A, 135648 to N.P.), Spanish Ministry of Science (Grant BFI-2014-57581-P to M.T.-S., co-funded with EU funds from FEDER Program), COST grant (Action BM1105), Deutsche Israel Program grant (DFG, to J.A. H.), the King's Bioscience Institute and the Guy's and St. Thomas' Charity Prize Ph.D. Programme in Biomedical and Translational Science (to E.J.L.), and Medical Research Council (MR/L016729/1 to C.L.A.).

## Author contributions

K.K., T.R., R.V., M.L.-N., R.K.-F., T.E., N.P. and F.P.-H. collected the study subjects and the clinical data. K.K., T.R., R.V., M.L.-N., N.P., F.P.-H., L.V., E.-M.S., J.K.K., S.K. and T.E. ascertained the subjects and phenotyped the patients. J.T., P.L. and M.K. performed the pedigree and linkage analyses. J.T., R.F., and T.R. analyzed the whole-genome sequencing data. J.T., J.K. and K.V. performed the Sanger-sequencing analyses. T.J., C.T., L.S., L.Y., K.V., S.V., K.P., T.R., J.K., J.T., M.T.-S. and M.L. performed the functional analyses and interpreted the results. T.B. and J.A.H. performed the KCNQ1 molecular modeling. X.L. and M.V. performed the proteomics analyses. E.J.L., C.L.A., P.M., J.K., P.G., F.C. and K.V. studied KCNQ1 expression. T.R. coordinated and managed the study. T.R., J.T., J.K., K.V., M.T.-S., T.J., M.V., C.L.A. and J.A.H. wrote the manuscript. All authors critically revised the manuscript.

## Additional information

**Competing interests:** The authors declare no competing financial interests.

Johanna Tommiska[1,2], Johanna Känsäkoski[1], Lasse Skibsbye[3], Kirsi Vaaralahti[1], Xiaonan Liu[4], Emily J. Lodge[5], Chuyi Tang[3], Lei Yuan[3], Rainer Fagerholm[1,6], Jørgen K. Kanters[7,8], Päivi Lahermo[9], Mari Kaunisto[9], Riikka Keski-Filppula[10], Sanna Vuoristo[1], Kristiina Pulli[1], Tapani Ebeling[11], Leena Valanne[12], Eeva-Marja Sankila[13], Sirpa Kivirikko[14], Mitja Lääperi[1], Filippo Casoni[15,16], Paolo Giacobini[15,16], Franziska Phan-Hug[17], Tal Buki[18], Manuel Tena-Sempere[19,20,21], Nelly Pitteloud[17], Riitta Veijola[22,23], Marita Lipsanen-Nyman[2], Kari Kaunisto[22], Patrice Mollard[24], Cynthia L. Andoniadou [5,25], Joel A. Hirsch[18], Markku Varjosalo[4], Thomas Jespersen[3] & Taneli Raivio[1,2]

[1]Faculty of Medicine, Department of Physiology, University of Helsinki, 00014 Helsinki, Finland. [2]Children's Hospital, Pediatric Research Center, Helsinki University Central Hospital (HUCH), 00029 Helsinki, Finland. [3]Department of Biomedical Sciences, University of Copenhagen, 2200 Copenhagen N, Denmark. [4]Institute of Biotechnology, Biocenter 3, University of Helsinki, 00014 Helsinki, Finland. [5]Centre for Craniofacial and Regenerative Biology, King's College London, Floor 27 Tower Wing, Guy's Campus, London SE1 9RT, UK. [6]Department of Obstetrics and Gynecology, HUCH, 00029 Helsinki, Finland. [7]Laboratory of Experimental Cardiology, Department of Biomedical Sciences, University of Copenhagen, 22000 Copenhagen, Denmark. [8]Department of Cardiology, Herlev & Gentofte University Hospitals, University of Copenhagen, 22000 Copenhagen, Denmark. [9]Institute for Molecular Medicine Finland (FIMM), Helsinki Institute of Life Science HiLIFE, University of Helsinki, 00014 Helsinki, Finland. [10]Department of Clinical Genetics, Oulu University Hospital, 90029 Oulu, Finland. [11]Department of Medicine, Oulu University Hospital, Finland and Research Unit of Internal Medicine, University of Oulu, 90014 Oulu, Finland. [12]Helsinki Medical Imaging Center, HUCH, 00029 Helsinki, Finland. [13]Department of Ophthalmology, HUCH, 00029 Helsinki, Finland. [14]Department of Clinical Genetics, HUCH, 00029 Helsinki, Finland. [15]Inserm U1172, Jean-Pierre Aubert Research Center, Development and Plasticity of the Neuroendocrine Brain, 59045 Lille, France. [16]University of Lille, School of Medicine, 59045 Lille, France. [17]Pediatrics, Division of Pediatric Endocrinology, Diabetology and Obesity, University Hospital Lausanne (CHUV), 1011 Lausanne, Switzerland. [18]Department of Biochemistry and Molecular Biology, George S. Wise Faculty of Life Sciences, Institute of Structural Biology, 69978 Ramat Aviv, Israel. [19]Department of Cell Biology, Physiology and Immunology, University of Córdoba, 14071 Cordoba, Spain. [20]Instituto Maimonides de Investigacion Biomedica (IMIBIC/HURS), 14004 Cordoba, Spain. [21]CIBER Fisiopatología de la Obesidad y Nutrición, Instituto de Salud Carlos III, 28029 Madrid, Spain. [22]Department of Children and Adolescents, Oulu University Hospital, 90029 Oulu, Finland. [23]Department of Pediatrics, PEDEGO Research Center, Medical Research Center, University of Oulu, 90014 Oulu, Finland. [24]IGF, CNRS, INSERM, Univ. Montpellier, F-34094 Montpellier, France. [25]Department of Internal Medicine III, Technische Universität Dresden, Fetscherstraße 74, 01307 Dresden, Germany. Johanna Tommiska and Johanna Känsäkoski contributed equally to this work.

