## [Peer review file · Nature Communications]

Reviewer #1 (Remarks to the Author):

Combining linkage analysis with whole-genome sequencing in patients with growth hormone deficiency and maternally inherited gingival fibromatosis, this work shows that patients from three unrelated families harbored either one of the two novel missense mutations, R116L or P369L, in the potassium channel KCNQ1 alpha subunit, whose defected gene was previously shown to be involved in cardiac arrhythmias such as the long QT or short QT syndromes and atrial fibrillation. Clinically, the diseased patients carrying either one of the two KCNQ1 mutations displayed a wide endocrine phenotypic spectrum that ranged from relatively mild (only low IGF-1 levels in 2 patients, or gingival fibromatosis in one individual) through classic growth hormone deficiency to multiple pituitary hormone deficiencies. The authors examined the electrophysiological consequences of the KCNQ1 mutants expressed either alone or in combination with the KCNE1 or KCNE2 auxiliary subunits. In most cases, the currents generated by the mutants were larger than those of the WT KCNQ1 channel and seemed to indicate gain-of-function mutations. Then, to relate the KCNQ1 mutations to growth hormone deficiency, the authors setup an Elisa immunoassay of pituitary hormone secretion by transiently overexpressing WT or one of the KCNQ1 mutants in ACTH-secreting mouse pituitary tumor cell line AtT-20. ACTH levels produced by cells transfected with the KCNQ1 mutants expressed alone did not differ from WT KCNQ1 but those induced by mutants co-expressed with KCNE2 were significantly lower compared to WT. The authors suggest that gain-of-function of the mutants may hyperpolarize pituitary cells and greatly diminish the release of ACTH.

This work is potentially interesting and explores a very important issue that relies on the pathophysiological mechanisms and defective genes involved in inherited growth hormone deficiency. However, the authors desperately seek for a possible mechanism or explanation that would account for the link between KCNQ1 mutations found in family members suffering from growth hormone deficiency. Unfortunately, the mechanisms or explanations raised by the authors are not clear, very vague and speculative. Below, I detail some of the major concerns:

1-The authors use the mouse pituitary tumor cell line AtT-20 to examine the impact of KCNQ1 mutations on ACTH secretion by overexpressing the mutants in absence or presence of KCNE subunits. This approach is not appropriate because it uses over-expression of the mutant channels in a pituitary tumor cell line and will therefore produce many biases. The experimental setup collects leaks of ACTH from AtT-20 cells with no context of stimulated secretion. Furthermore, the authors do not produce quantitative data (supplementary Figure 2) showing that the transfected cells express similar levels of WT KCNQ1 and mutant proteins as well as KCNE2 subunits. Instead, they provide one Western blot of very poor quality with no replicates and quantifications. They show that only in the presence of KCNE2, the KCNQ1 mutants produce lower ACTH release compared to WT. However, if one examines the currents obtained in HEK293 cells (Figure 2) and if one compares the current densities of KCNQ1 mutants R116L or P369L expressed alone and those obtained with the mutants+KCNE2, the differences, notably for R116L are not very large. So why, ACTH levels are lower in R116L+KCNE2 and not in R116L alone. Also, why the KCNQ1 mutant G589D, which is a loss-of-function mutation, yields the same ACTH levels compared to WT?

2-With the gain of function mutations, the affected individuals are expected to show clinical symptoms of atrial fibrillation. QTc values less than 350 ms are considered as abnormally short. In supplementary Table 1, the authors find that 5 patients exhibit shorter QTc intervals. In fact only 2 patients exhibit shorter QTc intervals according to the above criteria. These cardiac phenotypes are rather borderline and do not demonstrate firmly cardiac arrhythmias in the affected individuals.

3-In Figure 2c (middle panel), the current recording of transfected HEK293 cells are not performed in the appropriate way. The G/V relations show that no saturation of channel conductance is reached. The voltage step protocol should depolarize the WT KCNQ1 and its mutants+KCNE1 to larger depolarized potentials. Also, the authors in page 5 write: "the activation kinetics were shifted to more positive potentials" This sentence is not correct. Instead, the voltage dependence of activation is shifted to more depolarized potentials.

In all, the mechanisms raised by the authors are not clear, very vague and speculative. The study could reach a significant advance if knock-in mice carrying the KCNQ1 mutations could be produced and would exhibit similar growth hormone deficiency phenotype. At this stage, I suggest

the authors to submit their work to more specialized journals.

Reviewer #2 (Remarks to the Author):

Tommiska et al had identified a small group of patients with growth retardation due to GHD in combination with gingival fibromatosis for which a molecular diagnosis had not been given. Through linkage study of the largest pedigree they identify a heterozygous variant in KCNQ1 which co-segregate with the affected status in the pedigree. Two further families with same phenotype are screened for variants in KCNQ1 and the finding of another unreported heterozygous variant corroborates that rare variants in KCNQ1 could cause the give phenotype. To prove this they do various electrophysiological studies etc. and concludes that these mutations are indeed causing changes to the functioning of the KCNQ1 ion channel.

This is an interesting paper which shows that GHD w/ associated GF is an allelic disorder with long QT and short QT syndromes etc. and clearly adds to the complexity of disease that we see in other allelic disorders.

I have a few comments to the paper:

Page 5, line 131: you mention KCNE2. I am not totally familiar with the literature on KCNQ1 and other associated proteins, but I am wondering if referring to Roepke et al 2009 is warranted here.

Page 5, line 135: you refer to Fig 1c and sup fig 1 when talking about how the two mutations stabilizes the intracellular channel structure... That is not really what those figures show.

Page 6, line 173: you refer to maternal imprinting. However, you only have 3 families with 4 maternal GF vs 2 paternal non-GF. So you are definitely right in using the word 'suggesting' as you have too few samples to really be able to give a firm statement on this. You could help your point by discussing this a bit further so I would recommend you expand a bit on this here. You probably ought to refer to Fitzpatrick et al 2002 and Mancini-DiNardo et al 2003. I think the discussion around this is a bit weak the way it is presented here.

Page 6, line 177: you refer to cyclosporin by the way causes GF as a side effect... So what? I know what you are trying to say, but I think the statement is just hanging there in the air. Please elaborate a bit and help the reader to think through this point.

Page 11, figure 1a: This is a bit tricky as you have not labelled all the samples (which is not necessary) anyhow, female I,1 (generation 1, sample 1) is an obligate carrier. You need to indicate that with the right signature. You also claim in the text that the twins #7 and #8 are monozygotic. In the pedigree they appear to be dizygotic at the moment. If they really are monozygotic, then add a horizontal bar to indicate that.

Page 14, line 315: You say you used Human OmniExpress, that array has 700k markers. Did you use all those in your linkage analysis? Please elaborate.

Page 15, line 338: are you going to provide primer sequences in the suppl. Section?

Supplementary Figure 1: I'm not familiar with this type of plot and I think it would be great if you could elaborate a bit more on the interpretation of (b). All you say is that disorder prediction correlates well with the conservation. How can I see that from the given plot?

Reviewer #3 (Remarks to the Author):

Two missense mutations in KCNQ1 cause pituitary hormone deficiency and maternally inherited gingival fibromatosis

Tommiska et al.

Using a combination of linkage analysis and whole genome sequencing in a large family with an autosomal dominant condition of growth hormone deficiency and gingival hyperplasia, Tommiska et al. identified a novel missense variant in KCNQ1. Follow up studies in two unrelated families with a similar clinical presentation revealed a different rare KCNQ1 variant that segregated with disease or was de novo (in 1 individual). In vitro functional studies suggested the KCNQ1 variants identified (p.R116L and p.R369L) generally displayed a gain of function effect, with higher current responses when expressed alone or with KCNE1/2. Importantly, this contrasts with the established loss of function mechanism associated with KCNQ1 in Long QT syndrome. Characterization of the missense variants in a mouse pituitary tumor cell line demonstrated diminished ACTH levels, providing a functional link to the endocrine defects observed in patients. Interactome analysis attempted to identify altered protein interactions that may underlie the associated phenotypes, though significant alterations due to the missense variants in KCNQ1 were not observed.

The manuscript is well written and describes a very interesting new clinical entity associated with KCNQ1 mutations. The authors provide sufficient clinical and functional data to implicate KCNQ1 variants in this novel disorder and demonstrate a new mechanism of disease associated with KCNQ1. I believe this study would be of interest to a broad audience. With minor edits, this manuscript should be suitable for publication in Nature Communications.

The following comments

Please indicate which individuals in Pedigree I were subjected to WGS.

Fig 2b and 2c are not referred to in the text

It may be beneficial to move Supplemental fig 2/table 2 into main manuscript and display as a histogram. Define units of measurement in this figure and consider representing data as a % change relative to controls.

Figure 2: please define the abbreviations used.

Figure S2. Please provide densitometric comparisons of these samples.

It is interesting that some phenotypes identified in the families studied segregate in an autosomal dominant fashion, while gingival fibromatosis follows a segregation pattern consistent with imprinting. Can the authors provide data to show that this tissue or its precursors exhibit monoallelic expression while others do not?

Reviewers' comments:

Reviewer #1 (Remarks to the Author):

Combining linkage analysis with whole-genome sequencing in patients with growth hormone deficiency and maternally inherited gingival fibromatosis, this work shows that patients from three unrelated families harbored either one of the two novel missense mutations, R116L or P369L, in the potassium channel KCNQ1 alpha subunit, whose defected gene was previously shown to be involved in cardiac arrhythmias such as the long QT or short QT syndromes and atrial fibrillation. Clinically, the diseased patients carrying either one of the two KCNQ1 mutations displayed a wide endocrine phenotypic spectrum that ranged from relatively mild (only low IGF-1 levels in 2 patients, or gingival fibromatosis in one individual) through classic growth hormone deficiency to multiple pituitary hormone deficiencies. The authors examined the electrophysiological consequences of the KCNQ1 mutants expressed either alone or in combination with the KCNE1 or KCNE2 auxiliary subunits. In most cases, the currents generated by the mutants were larger than those of the WT KCNQ1 channel and seemed to indicate gain-of-function mutations. Then, to relate the KCNQ1 mutations to growth hormone deficiency, the authors setup an Elisa immunoassay of pituitary hormone secretion by transiently overexpressing WT or one of the KCNQ1 mutants in ACTH-secreting mouse pituitary tumor cell line AtT-20. ACTH levels produced by cells transfected with the KCNQ1 mutants expressed alone did not differ from WT KCNQ1 but those induced by mutants co-expressed with KCNE2 were significantly lower compared to WT. The authors suggest that gain-of-function of the mutants may hyperpolarize pituitary cells and greatly diminish the release of ACTH. This work is potentially interesting and explores a very important issue that relies on the pathophysiological mechanisms and defective genes involved in inherited growth hormone deficiency. However, the authors desperately seek for a possible mechanism or explanation that would account for the link between KCNQ1 mutations found in family members suffering from growth hormone deficiency. Unfortunately, the mechanisms or explanations raised by the authors are not clear, very vague and speculative.

We thank the reviewer for her/his comments. Please find our detailed responses to your comments below. Please also note that we now present new data which is relevant to the putative mechanism(s) of this disease. We show membranous KCNQ1 staining in mouse somatotropes, and co-expression of *Kcnq1* and *Gh* in somatotrope and *Kcnq1* and *Lhb* in gonadotrope cells by using dual *in situ* hybridization technique (Supplementary Fig. 3). Our new data also show colocalization of *Ghrh* and *Kcnq1* in hypothalamic neurons (Supplementary Fig 3). These findings suggest that the R116L and P369L *KCNQ1* mutations impact hypothalamic-pituitary function at multiple levels; in the case of the growth hormone axis, both on the control of episodic GHRH secretion by hypothalamic neurons as well as on somatotroph function during growth. Intriguingly, *Kcnq1* was also expressed along the blood vessels of the developing mouse pituitary, and KCNQ1 was immunostained along the postnatal pituitary capillaries (Supplementary Fig. 3). Somatotrophs are organized to the close vicinity of the pituitary blood vessels (Le Tissier *et al.* Nat Rev Endocrinol. 2017;13:257-267), and GHRH enhances oxygen supply to the somatotroph network via increased capillary blood flow (Lafont *et al.* Proc Natl Acad Sci U S A. 2010;107:4465-70; Le Tissier *et al.* Nat Rev Endocrinol. 2017;13:257-267). Thus, it is possible that the two *KCNQ1* mutations also impair hormone secretion by altering regional pituitary blood flow. We have now added these results to the amended version of the manuscript, and discuss them in the main text (lines 150-164).

While we were preparing our response to Nature Communications, the cryo-EM structure of KCNQ1/CaM complex was published in Cell on June 1st (Sun and MacKinnon, Cell 2017;169:1042-1050). With the help of the newly available structural data we have been able to

fully map the structural alterations brought about by both mutations (Fig. 1c). These timely new findings are now also discussed in the manuscript.

Below, I detail some of the major concerns:

1-The authors use the mouse pituitary tumor cell line AtT-20 to examine the impact of KCNQ1 mutations on ACTH secretion by overexpressing the mutants in absence or presence of KCNE subunits. This approach is not appropriate because it uses over-expression of the mutant channels in a pituitary tumor cell line and will therefore produce many biases. The experimental setup collects leaks of ACTH from AtT-20 cells with no context of stimulated secretion.

We agree that all methods of molecular biology are never completely bias-free. For example, even genetic engineering with the CRISPR/Cas9 system has been suggested to bear an unexpectedly high off-target rate (Schaefer *et al.* Nature Methods 2017;14:547-548). More traditional methods, such as transient overexpression of the protein of interest have limitations as well. However, this method is still widely used in articles published in the very latest issues of high impact journals, such as the Journal of Clinical Investigation (Teveroni E. *et al.* J Clin Invest. 2017;127:1531-1545), Nature Communications (Mauriac S. *et al.* Nat Commun. 2017;8:14907) and American Journal of Human Genetics (Arno G. *et al.* Am J Hum Genet. 2017;100:334-342). We also emphasize that we have used a variety of methods (in addition to transient overexpression) ranging from whole genome sequencing, molecular modeling, immunohistochemistry, dual ultra-sensitive RNAscope *in situ* hybridization and mass spectrometry-based protein-protein interaction detection.

The AtT-20 cell line was selected because it is well-characterized, and to the best of our knowledge, represents the only commonly available pituitary line that has retained responsiveness to exogenous hypothalamic stimulating hormones (in this case CRH) (McArthur *et al.* FASEB J 2009;23:4000-4010, Heisler S *et al.* PNAS 1982;79:6502-6506). This cell line has also been widely used to investigate the regulation of exocytosis by voltage-gated calcium entry (Loechner *et al.* Endocrinology. 1996;137:1429-1437, Wang & Greer. Mol. Cell. Endocrinol. 1995;109:11-18). Therefore, we consider that the selection of our cell line was appropriate and suitable for our work. We have now emphasized this in the manuscript (lines 169-170). Please note that we indeed have investigated the stimulated secretion; please refer to the supplementary note on statistical analysis of ACTH secretion studies in the original submission, where we describe that stimulated levels did not differ between the different conditions.

Furthermore, the authors do not produce quantitative data (supplementary Figure 2) showing that the transfected cells express similar levels of WT KCNQ1 and mutant proteins as well as KCNE2 subunits. Instead, they provide one Western blot of very poor quality with no replicates and quantifications.

We have elaborated this in the amended version of the manuscript, and quantitated the expression of WT and mutant KCNQ1s, as requested. Relative expression of the p.Pro369Leu did not differ from the WT KCNQ1, whereas the expression of p.Arg116Leu was lower than that of the WT KCNQ1 ($P < 0.05$) (Supplementary Fig. 5). The lowest relative expression was exhibited by the additional control (p.Gly589Asp) ($P < 0.005$ vs. WT KCNQ1). Importantly, however, the ACTH levels produced by the cells transfected with the KCNQ1-Gly589Asp and KCNE2 did not differ from the WT condition (Supplementary Note, Supplementary Table 2b), suggesting that the differences in relative KCNQ1 expression levels did not determine the ACTH levels measured in the culture medium. These findings are now reported in the manuscript (lines 177-186).

Figure R1. Quantification of WT and mutant KCNQ1 expression in AtT-20 cells. The western blots have been cropped for clarity. β -actin served as a protein loading control and was imaged from the same blots as KCNQ1. The mean (\pm SD) of three independent experiments are shown. * $p < 0.05$ **, $p < 0.005$ (t-test against WT KCNQ1, which was set to 1.00).

In our ACTH assay, we transfected AtT-20 cells with 250 ng of different KCNQ1s and a constant amount of (250 ng) of WT KCNE2. The expected KCNE2 band of ~22 kDa (Parkington et al. Nat Communications 2014;5:4108) can be detected (**Figure R2**).

Figure R2. Western blotting of KCNE2. The AtT-20 cell lysates were obtained from the ACTH secretion experiments and were also used in the quantification of WT and mutant KCNQ1 constructs. The samples were run on 4-20% gradient gels, blotted onto either nitrocellulose (a) or PVDF (b) membranes, and detected with 1:200 rabbit polyclonal anti-KCNE2 (Alomone Labs) antibody. Two PMT (photo-multiplier tube) voltages are shown for both membranes. Yellow arrows indicate the expected KCNE2 bands (Parkington et al. Nat Communications 2014;5:4108). NT: non-transfected AtT-20 cells. The KCNQ1/KCNE2 samples in panels a. and b. are from independent transfections from the assays.

They show that only in the presence of KCNE2, the KCNQ1 mutants produce lower ACTH release compared to WT. However, if one examines the currents obtained in HEK293 cells (Figure 2) and if one compares the current densities of KCNQ1 mutants R116L or P369L expressed alone and those obtained with the mutants+KCNE2, the differences, notably for R116L are not very large. So why, ACTH levels are lower in R116L+KCNE2 and not in R116L alone. Also, why the KCNQ1 mutant G589D, which is a loss-of-function mutation, yields the same ACTH levels compared to WT?

The relatively largest effect of the two KCNQ1 mutants with KCNE2 are found at potentials close to the resting membrane potential of the cells. Hence, it could be speculated that the membrane potential gets slightly hyperpolarized, which is expected to reduce the probability of spontaneous depolarisation. In the case of endocrine cells, such a decrease in activity should diminish accumulation of secreted hormones, and this was what we found in the AtT-20 cells. We agree that one has to be cautious in extrapolating results obtained by using human embryonic kidney cells (HEKs) to the results obtained with an endocrine cell line. We carefully selected the best possible endocrine cell line that would be relevant to our disease to test if the mutations alone or in combination with KCNE2 would cause any difference in hormone secretion. Therefore, one should not (and we did not) expect 100% concordant results obtained with HEK293 cells (electrophysiological responses) and those obtained by the AtT-20 mouse pituitary cell line (ACTH secretion). As explained above, patients with the long QT syndrome due to loss-of-function mutations in *KCNQ1*, such as G589D, do not have pituitary hormone secretion disturbances, which was accurately recapitulated by our *in vitro* ACTH secretion model, as pointed out by the Reviewer. Finally (and as clearly implicated in the manuscript), the exact mechanism of this disease remains unresolved, although we feel that we have made significant progress in showing that KCNQ1 is expressed in relevant cell types for this disease, and by showing the first possible molecular level mechanisms that could lead to this complex phenotype.

2-With the gain of function mutations, the affected individuals are expected to show clinical symptoms of atrial fibrillation. QTc values less than 350 ms are considered as abnormally short. In supplementary Table 1, the authors find that 5 patients exhibit shorter QTc intervals. In fact only 2 patients exhibit shorter QTc intervals according to the above criteria. These cardiac phenotypes are rather borderline and do not demonstrate firmly cardiac arrhythmias in the affected individuals.

We have given the numbers of subjects below the 2nd percentile for age and gender. However, we agree it is a good idea, as the reviewer points out, to explicitly state that two subjects also fulfill the criteria for diagnosis of short QT syndrome (QTc interval \leq 340 ms; Priori SG *et al.* ESC guidelines for the management of patients with ventricular arrhythmias and the prevention of sudden cardiac death - European Heart Journal 2015;36:2793-2867). Accordingly, we have now modified the text in the manuscript as follows: Accordingly, 5 of the 12 mutation carriers examined (42%, 95% confidence interval 20%-68%) displayed a corrected QT interval below the 2nd percentile of age- and gender-matched reference values. Two of them fulfilled the diagnostic criteria for short QT syndrome (**Supplementary table 1**)^{17,18,19}.

3-In Figure 2c (middle panel), the current recording of transfected HEK293 cells are not performed in the appropriate way. The G/V relations show that no saturation of channel conductance is reached. The voltage step protocol should depolarize the WT KCNQ1 and its mutants+KCNE1 to larger depolarized potentials.

We thank the reviewer for this comment. Although the current kinetics of the KCNQ1/KCNE1 complex is not the main theme of our paper, we have performed new investigations on the

relationship between the depolarization time and activation kinetics of KCNQ1/KCNE1. This was done, as the reviewer correctly pointed out, because we did not have full time-dependent activation of the KCNQ1/KCNE1 channel complex. Hence, we have repeated the experiments now using 6 second depolarizing steps instead of 2 second.

We first investigated how many seconds of depolarization are needed to reach full activation of the KCNQ1/KCNE1 channel complex (see Figure R3 below) in HEK293 cells. These data depict that the channels gradually increased the conducted current for 4-6 seconds (depending on the voltage potential), after which they provide a stable current.

Figure R3. Activation kinetics of KCNQ1/KCNE1 channels following long depolarizing pulses. *Left:* Depolarising step for 8 seconds to +20 mV. *Right:* Incremental depolarizing step from -80 mV to +60 mV for 6 seconds.

Consequently, we used 6 second depolarizing steps in a series of new experiments with KCNQ1/KCNE1 WT/R116L/P369L channels, generally otherwise performed similarly as explained in the manuscript (Figure R4).

KCNQ1/KCNE1

Figure R4. Electrophysiological results for WT, KCNQ1-Arg116Leu and KCNQ1-Pro369Leu mutant KCNQ1 proteins co-expressed with KCNE1. Recordings of currents measured during the voltage-clamp protocol in HEK293 cells transfected with cDNAs encoding wild-type (WT) or mutants (p.Arg116Leu, p.Pro369Leu) *Left:* Current-voltage relationships normalized for cell size. Peak current analysis was performed at the end of each voltage step (for KCNQ1/KCNE1 following a 6 s depolarizing step). *Right:* Normalized activation curves, measured 2-4 ms after

stepping to -30 mV, as a function of the prior voltage potential (for KCNQ1/KCNE1 following a 6 s depolarizing step). Mean \pm SEM values are shown. *P<.05, ***P<.001 vs. WT.

These results nicely reflect the original data presented in Figure 2c (middle panel) of the manuscript, with an increased current density for P369L and a shift in the voltage dependence of activation to more depolarized potentials for R116L. These new results with longer depolarization times now substitute the original 2 s KCNQ1/KCNE1 data. Stepping to even more depolarized potentials > +60 mV might also have been of biophysical interest, but this is simply not possible combined with the long depolarizing steps. In particular, the steps to positive potentials for several seconds has a profound effect on the seal quality of the HEK293 cells; the more positive depolarization potential, the faster the giga Ω seal deteriorates.

Also, the authors in page 5 write:” the activation kinetics were shifted to more positive potentials” This sentence is not correct. Instead, the voltage dependence of activation is shifted to more depolarized potentials.

We have now edited the manuscript, as suggested by the reviewer.

In all, the mechanisms raised by the authors are not clear, very vague and speculative. The study could reach a significant advance if knock-in mice carrying the KCNQ1 mutations could be produced and would exhibit similar growth hormone deficiency phenotype. At this stage, I suggest the authors to submit their work to more specialized journals.

We refer to our first paragraph of this response letter, in which we explain the main findings of our new data that bolsters the putative mechanisms between KCNQ1 and the hypothalamic-pituitary unit function. We have given a serious consideration of generating a knock-in animal model, but we decided not to pursue this pathway in this first report for two reasons. First, in the era of iPSCs, the value of animal models in disease modeling has been questioned (Dolmetsch & Geschwind Cell 2011;145:831-4). Second, and perhaps even more importantly, the knock-in approach has not been proven successful to explain the mechanism of another channelopathy associated with an endocrine phenotype (hyperinsulinism and type 2 diabetes) in patients with a point mutation in the ABCC8 gene (Huopio et al. Lancet 2003; 361:301-7; Shimomura et al. Diabetes; 2013;62:3797-806). Despite the possible shortcomings of this approach, we agree, however, that this avenue still remains a lucrative possibility to be explored in the future papers.

Reviewer #2 (Remarks to the Author):

Tommiska et al had identified a small group of patients with growth retardation due to GHD in combination with gingival fibromatosis for which a molecular diagnosis had not been given. Through linkage study of the largest pedigree they identify a heterozygous variant in KCNQ1 which co-segregate with the affected status in the pedigree. Two further families with same phenotype are screened for variants in KCNQ1 and the finding of another unreported heterozygous variant corroborates that rare variants in KCNQ1 could cause the give phenotype. To prove this they do various electrophysiological studies etc. and concludes that these mutations are indeed causing changes to the functioning of the KCNQ1 ion channel.

This is an interesting paper which shows that GHD w/ associated GF is an allelic disorder with long QT and short QT syndromes etc. and clearly adds to the complexity of disease that we see in other allelic disorders.

We thank the reviewer for his/her comments and encouragement. Please find our detailed responses to your comments below. Please also note that, in the amended version of the manuscript, we present new data. We show membranous KCNQ1 staining in mouse somatotropes, and co-expression of *Kcnq1* and *Gh* in somatotrope and *Kcnq1* and *Lhb* in gonadotrope cells by using dual *in situ* hybridization technique (Supplementary Fig. 3). Our new data also shows colocalization of *Ghrh* and *Kcnq1* in hypothalamic neurons (Supplementary Fig 3). These findings suggest that the R116L and P369L *KCNQ1* mutations impact hypothalamic-pituitary function at multiple levels; in the case of the growth hormone axis, both on the control of episodic GHRH secretion by hypothalamic neurons as well as on somatotroph function during growth. Intriguingly, *Kcnq1* was also expressed along the blood vessels of the developing mouse pituitary, and KCNQ1 was immunostained along the postnatal pituitary capillaries (Supplementary Fig. 3). Somatotrophs are organized to the close vicinity of the pituitary blood vessels (Le Tissier *et al.* Nat Rev Endocrinol. 2017;13:257-267), and GHRH enhances oxygen supply to the somatotroph network via increased capillary blood flow (Lafont *et al.* Proc Natl Acad Sci U S A. 2010;107:4465-70; Le Tissier *et al.* Nat Rev Endocrinol. 2017;13:257-267). Thus, it is possible that the two *KCNQ1* mutations also impair hormone secretion by altering regional pituitary blood flow. We have now added these results to the amended version of the manuscript, and discuss them in the main text (lines 150-164).

While we were preparing our response to Nature Communications, the cryo-EM structure of KCNQ1/CaM complex was published in Cell on June 1st (Sun and MacKinnon, Cell 2017;169:1042-1050). With the help of the newly available structural data we have been able to fully map the structural alterations brought about by both mutations (Fig. 1c). These timely new findings are now discussed in the manuscript. These timely new results are now also discussed in the manuscript.

I have a few comments to the paper:

Page 5, line 131: you mention KCNE2. I am not totally familiar with the literature on KCNQ1 and other associated proteins, but I am wondering if referring to Roepke et al 2009 is warranted here.

We thank the reviewer for this important reference. In the amended version of the manuscript we now cite Roepke *et al.* when we say that KCNE2 is expressed in a number of tissues together with KCNQ1.

Page 5, line 135: you refer to Fig 1c and sup fig 1 when talking about how the two mutations stabilizes the intracellular channel structure... That is not really what those figures show.

We have now modified Fig 1c and supplementary figure 1. Using the recently reported cryo EM

structure, we map the mutated residues and observe that they interact with or are found in the intracellular CT/CaM gating module. Hence, we have modified our presentation, describing the molecular structure of the channel complex so that the reader can better understand the potential structure-function implications of the mutations (second paragraph, p 4). We then describe how neither mutation disrupts the proximal CT/CaM module's integrity, i.e. CaM still binds to the channel CT (supplementary figure 1 and supplementary figure 2). We show this by molecular modeling, biochemical pull-down assays, and the protein-protein interaction screen by mass spectrometry. Based on these findings and the electrophysiological results, we suggest that the mutations “impart subtle but distinct conformational changes to this module that favor the open channel state.”(p 5)

Page 6, line 173: you refer to maternal imprinting. However, you only have 3 families with 4 maternal GF vs 2 paternal non-GF. So you are definitely right in using the word ‘suggesting’ as you have too few samples to really be able to give a firm statement on this. You could help your point by discussing this a bit further so I would recommend you expand a bit on this here. You probably ought to refer to Fitzpatrick et al 2002 and Mancini-DiNardo et al 2003. I think the discussion around this is a bit weak the way it is presented here.

We thank the reviewer for this comment. If one includes only those subjects who have molecular genetic diagnosis to this analysis, and decides that each parent can contribute only once to this estimate (although they may have multiple children), one indeed ends up with 4 maternal inheritance events and 2 paternal inheritance events, as suggested by the Reviewer. However, if one performs a comprehensive pedigree analysis (rather than includes only those with a DNA sample available), it is evident that there are 6 “maternal inheritance and GF present” events (and zero “maternal inheritance, no GF” events) and 3 “paternal inheritance, no GF” events (and zero “paternal inheritance and GF present” events) among these three pedigrees, providing that each parent could contribute only once to this estimate. This association between the parent of origin of the disease and the presence or absence of GF is statistically significant (Fisher's exact test, $P < 0.05$). If we take an even wider perspective, however, there are 17 persons who have inherited the disease from the mother, who all have GF. At the same time, there are 6 persons who have inherited the disease from the father and none of them has GF. This association between the parent of origin of the disease and the presence or absence of GF is statistically highly significant (Fisher's exact test, $P = 0.00001$). Based on the above reasoning, we feel that it is appropriate to leave the parent-of-origin concept in the manuscript. We thank the reviewer for the two highly interesting reference suggestions, which describe the function of the imprinting control center in mice. Please note, however, that our data do not suggest impaired function of the human imprinting control center in chromosome 11p15.

Page 6, line 177: you refer to cyclosporin by the way causes GF as a side effect... So what? I know what you are trying to say, but I think the statement is just hanging there in the air. Please elaborate a bit and help the reader to think through this point.

We have now elaborated this sentence. In the amended version of the manuscript we write: Of note, cyclosporine A has been proposed to activate KCNQ1 and this immunosuppressive agent also frequently causes GF as a side effect. Thus, it is possible that KCNQ1 activation plays also a role in drug-induced GF.

Page 11, figure 1a: This is a bit tricky as you have not labelled all the samples (which is not necessary) anyhow, female I,1 (generation 1, sample 1) is an obligate carrier. You need to indicate that with the right signature. You also claim in the text that the twins #7 and #8 are monozygotic. In the pedigree they appear to be dizygotic at the moment. If they really are

monozygotic, then add a horizontal bar to indicate that.

We have now modified the figure as suggested.

Page 14, line 315: You say you used Human OmniExpress, that array has 700k markers. Did you use all those in your linkage analysis? Please elaborate.

We used all the 700k markers included in OmniExpress in our linkage analysis with the Pseudomarker software. The software removes markers that are not informative in any family from the analysis, and it sets no requirements to prune the data beforehand.

Page 15, line 338: are you going to provide primer sequences in the suppl. Section?

We have now added the primer sequences as suggested.

Supplementary Figure 1: I'm not familiar with this type of plot and I think it would be great if you could elaborate a bit more on the interpretation of (b). All you say is that disorder prediction correlates well with the conservation. How can I see that from the given plot?

This question is no longer relevant as we have removed this figure and replaced it with a different one, as discussed above.

Reviewer #3 (Remarks to the Author):

Two missense mutations in KCNQ1 cause pituitary hormone deficiency and maternally inherited gingival fibromatosis

Tommiska et al.

Using a combination of linkage analysis and whole genome sequencing in a large family with an autosomal dominant condition of growth hormone deficiency and gingival hyperplasia, Tommiska et al. identified a novel missense variant in KCNQ1. Follow up studies in two unrelated families with a similar clinical presentation revealed a different rare KCNQ1 variant that segregated with disease or was de novo (in 1 individual). In vitro functional studies suggested the KCNQ1 variants identified (p.R116L and p.R369L) generally displayed a gain of function effect, with higher current responses when expressed alone or with KCNE1/2. Importantly, this contrasts with the established loss of function mechanism associated with KCNQ1 in Long QT syndrome. Characterization of the missense variants in a mouse pituitary tumor cell line demonstrated diminished ACTH levels, providing a functional link to the endocrine defects observed in patients. Interactome analysis attempted to identify altered protein interactions that may underlie the associated phenotypes, though significant alterations due to the missense variants in KCNQ1 were not observed.

The manuscript is well written and describes a very interesting new clinical entity associated with KCNQ1 mutations. The authors provide sufficient clinical and functional data to implicate KCNQ1 variants in this novel disorder and demonstrate a new mechanism of disease associated with KCNQ1. I believe this study would be of interest to a broad audience. With minor edits, this manuscript should be suitable for publication in Nature Communications.

We thank the reviewer for his/her comments and encouragement. Please find our detailed responses to your comments below. Please also note that, in the amended version of the manuscript, we present new data. We show membranous KCNQ1 staining in mouse somatotropes, and co-expression of *Kcnq1* and *Gh* in somatotrope and *Kcnq1* and *Lhb* in gonadotrope cells by using dual *in situ* hybridization technique (Supplementary Fig. 3). Our new data also shows colocalization of *Ghrh* and *Kcnq1* in hypothalamic neurons (Supplementary Fig 3). These findings suggest that the R116L and P369L *KCNQ1* mutations impact hypothalamic-pituitary function at multiple levels; in the case of the growth hormone axis, both on the control of episodic GHRH secretion by hypothalamic neurons as well as on somatotroph function during growth. Intriguingly, *Kcnq1* was also expressed along the blood vessels of the developing mouse pituitary, and KCNQ1 was immunostained along the postnatal pituitary capillaries (Supplementary Fig. 3). Somatotrophs are organized to the close vicinity of the pituitary blood vessels (Le Tissier *et al.* Nat Rev Endocrinol. 2017;13:257-267), and GHRH enhances oxygen supply to the somatotroph network via increased capillary blood flow (Lafont *et al.* Proc Natl Acad Sci U S A. 2010;107:4465-70; Le Tissier *et al.* Nat Rev Endocrinol. 2017;13:257-267). Thus, it is possible that the two *KCNQ1* mutations also impair hormone secretion by altering regional pituitary blood flow. We have now added these results to the amended version of the manuscript, and discuss them in the main text.

While we were preparing our response to Nature Communications, the cryo-EM structure of KCNQ1/CaM complex was published in Cell on June 1st (Sun and MacKinnon, Cell 2017;169:1042-1050). With the help of the newly available structural data we have been able to fully map the structural alterations brought about by both mutations (Fig. 1c). These timely new

findings are now discussed in the manuscript. These timely new results are now also discussed in the manuscript.

The following comments

Please indicate which individuals in Pedigree I were subjected to WGS.

We have now added this information in the Pedigree I.

Fig 2b and 2c are not referred to in the text

We have now referred to these in the text.

It may be beneficial to move Supplemental fig 2/table 2 into main manuscript and display as a histogram. Define units of measurement in this figure and consider representing data as a % change relative to controls.

The current way of expressing these data (*i.e.* with the mixed model analysis) takes into account the time between sample collection, which adds the validity of the model. Unfortunately this information cannot be packed into a single value or figure without losing this information. To preserve the original observations in the model, we decided to leave the Supplemental Table 2 as is.

Figure 2: please define the abbreviations used.

We have now defined the abbreviations.

Figure S2. Please provide densitometric comparisons of these samples.

We have elaborated this in the amended version of the manuscript, and quantitated the expression of WT and mutant KCNQ1s, as requested. Relative expression of the p.Pro369Leu did not differ from the WT KCNQ1, whereas the expression of p.Arg116Leu was lower than that of the WT KCNQ1 ($P < 0.05$) (Supplementary Fig. 5). The lowest relative expression was exhibited by the additional control (p.Gly589Asp) ($P < 0.005$ vs. WT KCNQ1). Importantly, however, the ACTH levels produced by the cells transfected with the KCNQ1-Gly589Asp and KCNE2 did not differ from the WT condition (Supplementary Note, Supplementary Table 2b), suggesting that the differences in relative KCNQ1 expression levels did not determine the ACTH levels measured in the culture medium. These findings are now reported in the manuscript (lines 177-186).

Figure R1. Quantification of WT and mutant KCNQ1 expression in AtT-20 cells. The Western blots have been cropped for clarity. β -actin served as a protein loading control and was imaged from the same blots as KCNQ1. The mean (\pm SD) of three independent experiments are shown. * $p < 0.05$ ** $p < 0.005$ (t-test against WT KCNQ1, which was set to 1.00).

It is interesting that some phenotypes identified in the families studied segregate in an autosomal dominant fashion, while gingival fibromatosis follows a segregation pattern consistent with imprinting. Can the authors provide data to show that this tissue or its precursors exhibit monoallelic expression while others do not?

Unfortunately, we do not have such data. The overall expression of KCNQ1 appears biallelic in adult gingiva (Zhu et al. Human Genetics 2007; 121:113-123), but we were unable to find any published data on children on this topic. In addition, detailed analyses on the expression of KCNQ1 in different gingival cell types is not available. We agree with the reviewer that it would be of particular interest to explore which gingival cell types in children express *KCNQ1*, and if the expression was monoallelic in all cell types or just in certain subtypes such as in the mesenchymal gingival stem cells (Xu et al. J Dent Res 2013; 92:825-32).

Reviewer #1 (Remarks to the Author):

While the investigators have identified rare variants in the KCNQ1 gene in patients with growth hormone deficiency (GHD) and maternally inherited gingival fibromatosis (GF), they provide no functional proof that these variants are in fact associated with GHD and/or GF. Association of a new gene (not previously associated functionally) with a disease requires unambiguous strong functional evidence. An individual exome may contain dozens of variants and studies suggesting that a variant affects protein function, including a patient with a de novo mutation, are not a proof of disease causation. Ideally, a knock-in animal model or a human iPSCs cellular model would be best shaped to address this question. However, in their rebuttal the authors raise concerns about these approaches, which have been proven very successful for hundreds of gene mutations in many diseases.

In their revised version, the authors bring interesting new data, which unfortunately raise more concerns than addressing critical issues of the functional link to GHD.

1- The authors claim to bring new data relevant to the putative mechanism. They show by using dual in situ hybridization technique that membranous KCNQ1 staining in mouse somatotropes, and co-expression of KCNQ1 and GH in somatotrope and KCNQ1 and LH in gonadotrope cells. They also show colocalization of GHRH and KCNQ1 in hypothalamic neurons. They prematurely conclude that the R116L and P369L KCNQ1 mutations impact on hypothalamic-pituitary function at multiple levels; in the case of the growth hormone axis, both on the control of episodic GHRH secretion by hypothalamic neurons. If indeed the mutations significantly increased (gain-of-function) the K⁺ current only in the presence of KCNE2, as shown in Figure 2, then it is crucial that the authors should demonstrate the presence of KCNE2 mRNA or protein in somatotrope and gonadotrope cells as well as in hypothalamic neurons. According to the author's data, the presence of KCNE2 is decisive to confer the pathophysiological impact on the homeostasis of neurohormone secretion of the HP axis. Intriguingly, KCNE2 deletion does confer any HP axis hormone secretion phenotype (Roepke et al, Nat Med. 2009 (10):1186-94; Hu et al, Circ Cardiovasc Genet. 2014 (1):33-42).

2- The authors found that KCNQ1 is also expressed along the blood vessels of the developing mouse pituitary, and KCNQ1 was immunostained along the postnatal pituitary capillaries. Thus, they suggest that the two KCNQ1 mutations also impair hormone secretion by altering regional pituitary blood flow. Indeed, GHRH enhances oxygen supply to the somatotroph network via increased capillary blood flow. However, according to the electrophysiological data, the two mutations are gain-of-function in the presence of KCNE2 and in this case, they will produce vasodilation in the capillaries and will increase the blood flow, thereby producing the same effect as GHRH. The group of Iain Greenwood (St George's, University of London) showed that KCNQ channels and notably KCNQ1 are expressed in smooth muscle cells of blood vessels and their activation has a vasorelaxant effect (Chadha et al, Br J Pharmacol. 2012 (4):1377-87; Khanamiri et al, Hypertension. 2013 (6):1090-7; Stott et al, Drug Discov Today. 2014 (4):413-24).

Altogether these observations are inconsistent with the author's assumption.

3-The authors indicate that two of the mutations carriers (subject 17 and 20, supplementary table 1) fulfilled the diagnostic criteria for short QT syndrome. However, in subject 17 the mutation R116L co-expressed with KCNE1 yields currents which are similar to those of WT KCNQ1+KCNE1, while in subject 20, the mutation P369L co-expressed with KCNE1 produced larger currents than those WT KCNQ1+KCNE1. Accordingly, we should expect a short QT interval only in patient 20 carrying the mutation P369L and not in patient 17 bearing the mutation R116L.

4-The authors suggest that the gain-of-function of the mutants may hyperpolarize pituitary cells and greatly diminish the release of ACTH. If the AtT-20 cell line is indeed a relevant cell model for GHD, it should be very important for the authors to check whether KCNQ1 and KCNE2 mRNAs or proteins are expressed in these cells and if so, whether siRNA directed against KCNQ1 will increase the secretion of ACTH. This approach would more consistent to the hypothesis and less invasive compared to overexpression of the KCNQ1 and KCNE2 genes.

5- In figure 2a, the authors should label the WT and the mutants as they did in the first version of the manuscript. In Figure 2c, at least for KCNQ1 and KCNQ1+KCNE1, they should fit the conductance-voltage relations by a Boltzmann function and provide the V₅₀ and slope values of the fits. In page 5, line 138, the authors should replace the sentence: "the activation kinetics were

shifted to more depolarized potentials” by “The voltage dependence of activation was shifted to more depolarized potentials”. No time constants for activation kinetics are provided. The relevant parameters that should be deduced from the Boltzmann fits should the V50 and slope values. In conclusion, while the authors made efforts to add data and tried to provide arguments for a potential mechanism, I found unfortunately, that they adopt a kind of circular reasoning, which leaves us with no rational and consistent explanations accounting for the pathological phenotype. A clear link between the KCNQ1 variants and the functional consequences on GHD is really missing. In this particular study, I think that producing iPSC from one of these patients should not be an insurmountable task and would be very valuable for exploring the link mechanisms underlying the KCNQ1 variants with GHD.

Reviewer #2 (Remarks to the Author):

I have read the revised manuscript and looked at the response letter and I am happy with the changes made to the manuscript, which has clearly strengthen the story.

Reviewer #3 (Remarks to the Author):

The authors have adequately addressed the minor concerns I raised as part of the first review.

Reviewer #1 (Remarks to the Author):

While the investigators have identified rare variants in the KCNQ1 gene in patients with growth hormone deficiency (GHD) and maternally inherited gingival fibromatosis (GF), they provide no functional proof that these variants are in fact associated with GHD and/or GF. Association of a new gene (not previously associated functionally) with a disease requires unambiguous strong functional evidence. An individual exome may contain dozens of variants and studies suggesting that a variant affects protein function, including a patient with a de novo mutation, are not a proof of disease causation. Ideally, a knock-in animal model or a human iPSCs cellular model would be best shaped to address this question. However, in their rebuttal the authors raise concerns about these approaches, which have been proven very successful for hundreds of gene mutations in many diseases. In their revised version, the authors bring interesting new data, which unfortunately raise more concerns than addressing critical issues of the functional link to GHD.

We thank reviewer #1 for his/her comments. To better appreciate the previous important contributions in the field, we have now cited three seminal papers on the role of ion channels in pituitary function (refs. 23, 24 and 50 in the manuscript), an excellent review on the possible role of Kv channels in the regulation of vascular tone (ref. 31) and the comprehensive and educational review on the role of KCNE2 (ref. 22) in the amended version of the manuscript.

We apologize for any ambiguity in the original text. Please note that we have performed linkage analysis, whole-genome sequencing, and segregation analysis to identify the causal mutation. The *de novo* mutation, identified in a patient born to unaffected parents, in fact provides very strong support for the causality of this mutation. We have not used exome sequencing in this work.

As discussed in our previous response to Reviewer #1, we agree that disease modeling based on the use of pluripotent stem cells and animal work are lucrative next steps to further dissect the exact disease mechanism at molecular level. We do not share the Reviewer's concerns about our new data, and address his/her concerns in a point-by-point fashion below.

1- The authors claim to bring new data relevant to the putative mechanism. They show by using dual in situ hybridization technique that membranous KCNQ1 staining in mouse somatotropes, and co-expression of KCNQ1 and GH in somatotrope and KCNQ1 and LH in gonadotrope cells. They also show colocalization of GHRH and KCNQ1 in hypothalamic neurons. They prematurely conclude that the R116L and P369L KCNQ1 mutations impact on hypothalamic-pituitary function at multiple levels; in the case of the growth hormone axis, both on the control of episodic GHRH secretion by hypothalamic neurons. If indeed the mutations significantly increased (gain-of-function) the K⁺ current only in the presence of KCNE2, as shown in Figure 2, then it is crucial that the authors should demonstrate the presence of KCNE2 mRNA or protein in somatotrope and gonadotrope cells as well as in hypothalamic neurons. According to the author's data, the presence of KCNE2 is decisive to confer the pathophysiological impact on the homeostasis of neurohormone secretion of the HP axis. Intriguingly, KCNE2 deletion does confer any HP axis hormone secretion phenotype (Roepke et al, Nat Med. 2009 (10):1186-94; Hu et al, Circ Cardiovasc Genet. 2014 (1):33-42).

Thank you for the comment. We showed that *Kcnq1* is being expressed in the hypothalamic GHRH neurons and also in pituitary somatotropes, both of which are the key cell types that regulate growth. We think that this justifies one to speculate that the two specific mutations in *KCNQ1*, R116L and P369L, may impact hypothalamic-pituitary function on multiple levels, but we have now toned-down the wording (page 5 and 6, lines 159-163). We agree that the results obtained with KCNQ1-KCNE2 complexes are an important lead on the way to complete understanding of the ultimate disease mechanisms. Therefore we investigated, as suggested by the Reviewer, if *KCNE2* is expressed in human tissues relevant for this disease. We now show that *KCNE2* is expressed in human pituitary and hypothalamus (Supplementary Figure 4c). Our new data are in agreement with

the expression of *Kcne2* in rodent hypothalamus (Tinel N. *et al.* FEBS Lett. 2000;480:137-41) and pituitary (Kim HJ. *et al.* J Endocrinol 2011;210:309-321). We have now presented these findings in the amended version of the manuscript (page 7, lines 195-204).

We also agree that the rapidly emerging endocrine manifestations of mutations in genes encoding ion channels could indeed be emphasized more in the manuscript. *Kcne2* *-/-* mice are not, however, expected to mimic the situation brought about by our two activating *KCNQ1* missense mutations in terms of function of the hypothalamic-pituitary (HP) axis, considering these are null and not gain-of-function mutants. In addition, the promiscuity of KCNE2 makes defining its exact physiological roles and the molecular etiology of its associated disease states quite challenging: KCNQ1, hERG1, Kv1.4, Kv1.5, Kv2.1, Kv3.1, and Kv3.2 are all known to be regulated by the KCNE2 (Kanda *et al.* Biophys J 2011;101:1354-1363, Kanda *et al.* Biophys J 2011;101:1364-1375, Lewis *et al.* J Biol Chem. 2004;279:7884-7892, McCrossan *et al.* J Membr Biol. 2009; 228:1-14, Roepke *et al.* FASEB J. 2008;22:3648-3660, Tinel *et al.* Embo J 2000;19:6326-6330). We agree that reports on pituitary function in *Kcne2* *-/-* mice have not yet been disclosed. In spite of the above limitations, very recent evidence shows that *Kcne2* indeed is implicated in hormone secretion, as *Kcne2* *-/-* mice display impaired insulin secretion, and these effects are speculated to be mediated by KCNQ1-KCNE2 complexes (Lee *et al.* FASEB J 2017;31:2674-2685). We discuss these findings briefly in the amended version of the manuscript and have further toned-down the text to avoid any impression that KCNE2 would be the sole key to the disease mechanism (page 7, lines 195-204).

2- The authors found that KCNQ1 is also expressed along the blood vessels of the developing mouse pituitary, and KCNQ1 was immunostained along the postnatal pituitary capillaries. Thus, they suggest that the two KCNQ1 mutations also impair hormone secretion by altering regional pituitary blood flow. Indeed, GHRH enhances oxygen supply to the somatotroph network via increased capillary blood flow. However, according to the electrophysiological data, the two mutations are gain-of-function in the presence of KCNE2 and in this case, they will produce vasodilation in the capillaries and will increase the blood flow, thereby producing the same effect as GHRH. The group of Iain Greenwood (St George's, University of London) showed that KCNQ channels and notably KCNQ1 are expressed in smooth muscle cells of blood vessels and their activation has a vasorelaxant effect (Chadha *et al.*, Br J Pharmacol. 2012 (4):1377-87; Khanamiri *et al.*, Hypertension. 2013 (6):1090-7; Stott *et al.*, Drug Discov Today. 2014 (4):413-24). Altogether these observations are inconsistent with the author's assumption.

We thank the Reviewer for these comments. We have now underscored the possible role of Kv channels in vascular physiology (ref. 31). We are, however, reluctant to cite the two other papers (by Chadha and Khanamiri, respectively) mentioned by the Reviewer, because R-L3, the Kv7.1 channel activator employed in those works, was quite recently suggested to be an inappropriate pharmacological tool for studying the function of native vascular Kv7.1 channels in mice (Tsvetkov *et al.* British Journal of Pharmacology 2017; 174:150-162). Due to the apparent controversies in the field, we decided to retain our original text in which we very briefly speculate the role of KCNQ1, if any, in the regulation of pituitary blood flow (page 6, lines 169-172). The anterior pituitary is indeed richly vascularized by fenestrated capillaries emanating from the pituitary portal system, and a direct arterial supply has also been demonstrated in human pituitary adenomas (Turner *et al.* Endocr Rev 2003;24:600-632; Radacot *et al.* Bull Assoc Anat (Nancy) 1986;70:5-12; Turner *et al.* J Clin Endocrinol Metab 2000;85:1159-62). These findings have now been added to the amended version of the manuscript (page 6, lines 167-169).

3-The authors indicate that two of the mutations carriers (subject 17 and 20, supplementary table 1) fulfilled the diagnostic criteria for short QT syndrome. However, in subject 17 the mutation R116L co-expressed with KCNE1 yields currents which are similar to those of WT KCNQ1+KCNE1, while in subject 20, the mutation P369L co-expressed with KCNE1 produced larger currents than those WT KCNQ1+KCNE1. Accordingly, we should expect a

short QT interval only in patient 20 carrying the mutation P369L and not in patient 17 bearing the mutation R116L.

The two generally accepted themes in genetics, incomplete penetrance and variable expressivity, need to be appreciated here. In the cardiac context, the incomplete penetrance means that not all patients carrying a mutation (R116L or P369L) necessarily have the QT interval phenotype. Variable expressivity means that if they do have the phenotype, it is variable. These two themes are very well-known in the context of inherited arrhythmias and among those with the long QT syndrome due to mutations in *KCNQ1* (Giudicessi & Ackerman. *Transl Res.* 2013;161:1-14). Against this background, it is expected to observe differences between *in vitro* results and patients' phenotypes. The concepts of incomplete penetrance and variable expressivity with regard to *KCNQ1* mutations have now been clarified in the manuscript (page 5, lines 142-144).

4-The authors suggest that the gain-of-function of the mutants may hyperpolarize pituitary cells and greatly diminish the release of ACTH. If the AtT-20 cell line is indeed a relevant cell model for GHD, it should be very important for the authors to check whether KCNQ1 and KCNE2 mRNAs or proteins are expressed in these cells and if so, whether siRNA directed against KCNQ1 will increase the secretion of ACTH. This approach would more consistent to the hypothesis and less invasive compared to overexpression of the KCNQ1 and KCNE2 genes.

We thank the reviewer for this comment. However, the new suggested experiments are not going to resolve the question about the disease mechanism, because silencing of *KCNQ1* with siRNAs, thus mimicking a loss-of-function mutation, in any given cell line is not an appropriate way to study the mechanisms of a disease caused by two gain-of-function mutations in *KCNQ1*. Please note that we have already captured the essentials of this argument by including the G589D mutant (loss-of-function *KCNQ1* mutation) as an additional control in our ACTH secretion assay.

5- In figure 2a, the authors should label the WT and the mutants as they did in the first version of the manuscript. In Figure 2c, at least for KCNQ1 and KCNQ1+KCNE1, they should fit the conductance-voltage relations by a Boltzmann function and provide the V50 and slope values of the fits. In page 5, line 138, the authors should replace the sentence: “the activation kinetics were shifted to more depolarized potentials” by “The voltage dependence of activation was shifted to more depolarized potentials”. No time constants for activation kinetics are provided. The relevant parameters that should be deduced from the Boltzmann fits should the V50 and slope values.

We agree with the reviewer and have now included Boltzmann fits and listed the requested values for *KCNQ1* and *KCNQ1/KCNE1* currents in Figure 2 (panel C and Figure legend, respectively). In the manuscript, we have now changed the wording as suggested (“The voltage dependence of activation was shifted to more depolarized potentials”) (page 5, line 138).

In conclusion, while the authors made efforts to add data and tried to provide arguments for a potential mechanism, I found unfortunately, that they adopt a kind of circular reasoning, which leaves us with no rational and consistent explanations accounting for the pathological phenotype. A clear link between the KCNQ1 variants and the functional consequences on GHD is really missing. In this particular study, I think that producing iPSC from one of these patients should not be an insurmountable task and would be very valuable for exploring the link mechanisms underlying the KCNQ1 variants with GHD.

We do not share the Reviewer's rather pessimistic view. Our view on the amount of work required for the generation of high-quality human stem cell-based disease models differs from the Reviewer's. For example, it took several years to generate and validate a protocol for the differentiation of hypothalamic GnRH neurons from hESCs and iPSCs (Lund C. *et al.* *Stem Cell Reports* 2016;7:149-5). We agree, however, that modeling the relevant tissues for this disease with hESCs and iPSCs will be a very appealing theme for the forthcoming years.

Reviewer #2 (Remarks to the Author):

I have read the revised manuscript and looked at the response letter and I am happy with the changes made to the manuscript, which has clearly strengthen the story.

We thank Reviewer #2 for these kind words, and the comments that significantly improved our manuscript. We also thank the Reviewer for appreciating the new data that we added to the manuscript in order to strengthen the relationship between the mutation and function of the hypothalamic-pituitary axis.

Reviewer #3 (Remarks to the Author):

The authors have adequately addressed the minor concerns I raised as part of the first review.

We thank Reviewer #3 for these kind words, and also for his/her focused comments along the review process that certainly improved the manuscript.

Reviewer #1 (Remarks to the Author)

While the investigators have identified rare variants in the KCNQ1 gene in patients with growth hormone deficiency (GHD) and maternally inherited gingival fibromatosis (GF), they provide no functional proof that these variants are in fact associated with GHD and/or GF. Association of a new gene (not previously associated functionally) with a disease requires unambiguous strong functional evidence. An individual exome may contain dozens of variants and studies suggesting that a variant affects protein function, including a patient with a de novo mutation, are not a proof of disease causation. Ideally, a knock-in animal model or a human iPSCs cellular model would be best shaped to address this question. However, in their rebuttal the authors raise concerns about these approaches, which have been proven very successful for hundreds of gene mutations in many diseases.

In their revised version, the authors bring interesting new data, which unfortunately raise more concerns than addressing critical issues of the functional link to GHD.

1- The authors claim to bring new data relevant to the putative mechanism. They show by using dual in situ hybridization technique that membranous KCNQ1 staining in mouse somatotropes, and co-expression of KCNQ1 and GH in somatotrope and KCNQ1 and LH in gonadotrope cells. They also show colocalization of GHRH and KCNQ1 in hypothalamic neurons. They prematurely conclude that the R116L and P369L KCNQ1 mutations impact on hypothalamic-pituitary function at multiple levels; in the case of the growth hormone axis, both on the control of episodic GHRH secretion by hypothalamic neurons. If indeed the mutations significantly increased (gain-of-function) the K⁺ current only in the presence of KCNE2, as shown in Figure 2, then it is crucial that the authors should demonstrate the presence of KCNE2 mRNA or protein in somatotrope and gonadotrope cells as well as in hypothalamic neurons. According to the author's data, the presence of KCNE2 is decisive to confer the pathophysiological impact on the homeostasis of neurohormone secretion of the HP axis. Intriguingly, KCNE2 deletion does confer any HP axis hormone secretion phenotype (Roepke et al, *Nat Med.* 2009 (10):1186-94; Hu et al, *Circ Cardiovasc Genet.* 2014 (1):33-42).

2- The authors found that KCNQ1 is also expressed along the blood vessels of the developing mouse pituitary, and KCNQ1 was immunostained along the postnatal pituitary capillaries. Thus, they suggest that the two KCNQ1 mutations also impair hormone secretion by altering regional pituitary blood flow. Indeed, GHRH enhances oxygen supply to the somatotroph network via increased capillary blood flow. However, according to the electrophysiological data, the two mutations are gain-of-function in the presence of KCNE2 and in this case, they will produce vasodilation in the capillaries and will increase the blood flow, thereby producing the same effect as GHRH. The group of Iain Greenwood (St George's, University of London) showed that KCNQ channels and notably KCNQ1 are expressed in smooth muscle cells of blood vessels and their activation has a vasorelaxant effect (Chadha et al, *Br J Pharmacol.* 2012 (4):1377-87; Khanamiri et al, *Hypertension.* 2013 (6):1090-7; Stott et al, *Drug Discov Today.* 2014 (4):413-24). Altogether these observations are inconsistent with the author's assumption.

3-The authors indicate that two of the mutations carriers (subject 17 and 20, supplementary table 1) fulfilled the diagnostic criteria for short QT syndrome. However, in subject 17 the mutation R116L co-expressed with KCNE1 yields currents which are similar to those of WT KCNQ1+KCNE1, while in subject 20, the mutation P369L co-expressed with KCNE1 produced larger currents than those WT KCNQ1+KCNE1. Accordingly, we should expect a short QT interval only in patient 20 carrying the mutation P369L and not in patient 17 bearing the mutation R116L.

4-The authors suggest that the gain-of-function of the mutants may hyperpolarize pituitary cells and greatly diminish the release of ACTH. If the AtT-20 cell line is indeed a relevant cell model for GHD, it should be very important for the authors to check whether KCNQ1 and KCNE2 mRNAs or proteins are expressed in these cells and if so, whether siRNA directed against KCNQ1 will increase

the secretion of ACTH. This approach would more consistent to the hypothesis and less invasive compared to overexpression of the KCNQ1 and KCNE2 genes.

5- In figure 2a, the authors should label the WT and the mutants as they did in the first version of the manuscript. In Figure 2c, at least for KCNQ1 and KCNQ1+KCNE1, they should fit the conductance-voltage relations by a Boltzmann function and provide the V50 and slope values of the fits. In page 5, line 138, the authors should replace the sentence: "the activation kinetics were shifted to more depolarized potentials" by "The voltage dependence of activation was shifted to more depolarized potentials". No time constants for activation kinetics are provided. The relevant parameters that should be deduced from the Boltzmann fits should the V50 and slope values.

In conclusion, while the authors made efforts to add data and tried to provide arguments for a potential mechanism, I found unfortunately, that they adopt a kind of circular reasoning, which leaves us with no rational and consistent explanations accounting for the pathological phenotype. A clear link between the KCNQ1 variants and the functional consequences on GHD is really missing. In this particular study, I think that producing iPSC from one of these patients should not be an insurmountable task and would be very valuable for exploring the link mechanisms underlying the KCNQ1 variants with GHD.

Reviewer #2 (Remarks to the Author)

I have read the revised manuscript and looked at the response letter and I am happy with the changes made to the manuscript, which has clearly strengthen the story.

Reviewer #3 (Remarks to the Author)

REVIEWERS' COMMENTS:

Reviewer #1 (Remarks to the Author):

I appreciate the efforts made by the authors to address my concerns, especially in Figure 4C by checking the expression of KCNE2 in pituitary and hypothalamus. I think that the manuscript has been improved regarding some misinterpretations of the conductance measurements and other issues (Figure 2). Overall and although I do not agree with several arguments provided by the authors, the MS has been improved enough to be published in its current form.

We thank the reviewer for his/her kind words and for his/her thorough expert review of our manuscript. We believe that the Reviewer's comments have helped improve our manuscript, and although he/she does not fully agree with all of our arguments, we are happy to hear that the Reviewer now finds our manuscript suitable for publication.